# Visual Instruction Inversion:
# Image Editing via Visual Prompting

**Thao Nguyen    Yuheng Li    Utkarsh Ojha    Yong Jae Lee**
University of Wisconsin-Madison

`https://thaoshibe.github.io/visii/`

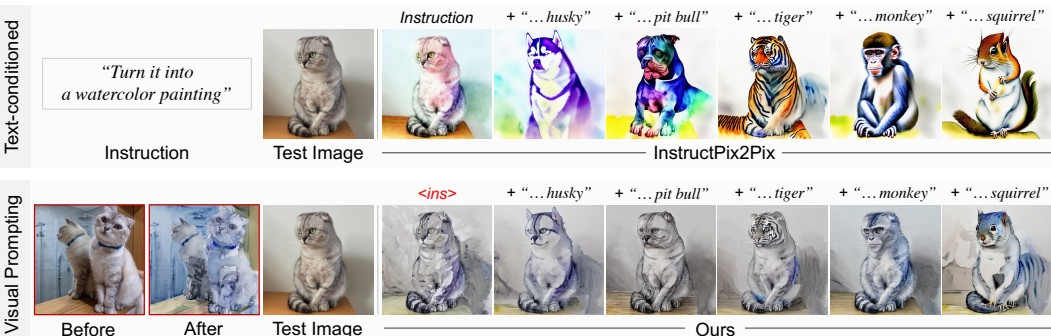

Figure 1: **Image editing via visual prompting.** Given a pair of *before-and-after images* of an edit, our approach (bottom) can *learn and apply* that edit along with the user's text prompt to enable a more accurate and intuitive image editing process compared to text-only conditioned approaches (top).

## Abstract

Text-conditioned image editing has emerged as a powerful tool for editing images. However, in many situations, language can be ambiguous and ineffective in describing specific image edits. When faced with such challenges, visual prompts can be a more informative and intuitive way to convey the desired edit. We present a method for image editing via visual prompting. Given example pairs that represent the "before" and "after" images of an edit, our approach learns a text-based editing direction that can be used to perform the same edit on new images. We leverage the rich, pretrained editing capabilities of text-to-image diffusion models by inverting visual prompts into editing instructions. Our results show that even with just one example pair, we can achieve competitive results compared to state-of-the-art text-conditioned image editing frameworks.

## 1   Introduction

In the past few years, diffusion models [37, 38, 6, 29, 40, 8] have emerged as a powerful framework for image generation. In particular, text-to-image diffusion models can generate stunning images conditioned on a text prompt. Such models have also been developed for *image editing* [27, 4, 19, 30, 54, 13, 16, 39, 9, 26]; i.e., transforming an image into another based on a text specification. As these models rely on textual guidance, significant effort has been made in prompt engineering [49, 12, 48], which aims to find well-designed prompts for text-to-image generation and editing.

But *what if the desired edit is difficult to describe in words?* For example, describing *your* drawing style of *your* cat can be challenging to put into a sentence (Fig. 2a). Or imagine that you want to transform a roadmap image into an aerial one – it could be difficult to know what the different colored

37th Conference on Neural Information Processing Systems (NeurIPS 2023).

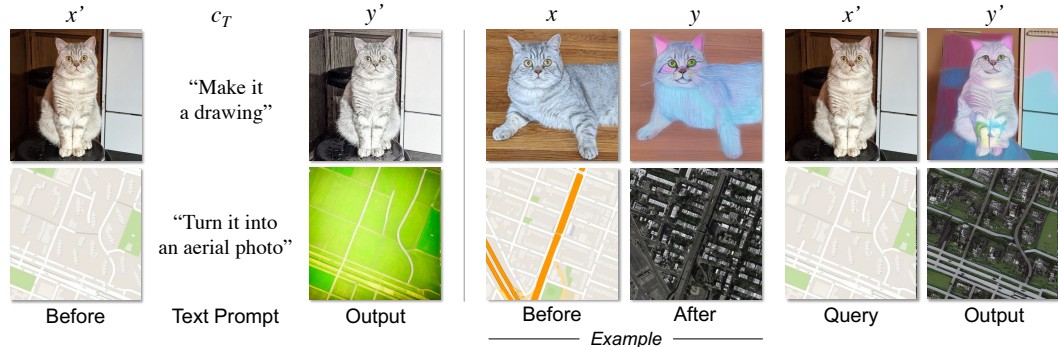

| $x'$ | $c_T$ | $y'$ | $x$ | $y$ | $x'$ | $y'$ |

"Make it a drawing"

"Turn it into an aerial photo"

| Before | Text Prompt | Output | Before | After | Query | Output |

*— Example —*

(a) Prior Work: Text-conditioned image editing          (b) Our approach: "Visual prompting" image editing

Figure 2: **Image editing with visual prompting.** (a) Text-conditioned scheme (Prior work): Model takes an input image and a text prompt to perform the desired edit. (b) Visual prompting scheme (Ours): Given a pair of before-after images of an edit, we learn an implicit text-based editing instruction, and apply it to new images.

regions in the roadmap image are supposed to represent, leading to an incorrect output image. In such cases, it would be easier, and more direct, to convey the edit *visually* by showing a before-and-after example pair (Fig. 2b). In other words, language can be ambiguous when describing a specific image edit transformation, while visual prompts can offer a more intuitive and precise way to describe it.

Visual prompting for image editing has very recently been explored in [3, 44]. These works reformulate the problem as an image in-painting task, where an example image pair ("before" and "after") and query image are provided in a single grid-like image. The target output is inpainted (by forming the analogy, `before:after = query:output`). After training on a large dataset of computer vision tasks (e.g., edge detection, bounding box localization), these systems aim to perform any of those tasks during testing with in-context learning [3, 44–46] without further fine-tuning. However, while they can work reasonably well for standard computer vision tasks such as segmentation and colorization, they cannot be used for general image editing tasks since large datasets for arbitrary edits are typically unavailable.

This paper investigates image editing via visual prompting using text-to-image diffusion models. Inspired by textual inversion [10], which inverts a visual identity specified by an example image into the rich, pre-trained text embedding of a large vision-and-language model [38, 32] for text-to-image generation, we propose to *invert the visual edit transformation specified by the example before-and-after image pair into a text instruction*. In particular, we leverage the textual instruction space that InstructPix2Pix [4] has learned. Since InstructPix2Pix directly builds upon a pretrained Stable Diffusion model's vast text-to-image generation capabilities, while further finetuning it with 450,000 (`text instruction, before image, after image`) triplets, our hypothesis is that its learned instruction space is rich enough to cover many image-to-image translations (i.e., image edits), and thus can be fruitful for visual prompt based image editing. Specifically, given a pair of images representing the "before" and "after" states of an editing task, we learn the edit direction in text space by optimizing for the textual instruction that converts the "before" image into the "after" image. Once learned, this edit direction can then be applied to a new test image, together with a text prompt, facilitating precise image editing; see Figure 1.

Our contributions and main findings are: (1) We introduce a new scheme for image editing via visual prompting. (2) We propose a framework for inverting visual prompts into editing instructions for text-to-image diffusion models. (3) By conducting in-depth analyses, we share valuable insights about image editing with diffusion models; e.g., concatenating instructions between learned and natural language yields a hybrid editing instruction that is more precise; or reusing the same noise schedule in training for testing leads to a more balanced result between editing effects and faithfulness to the input image.

## 2   Related Work

**Text-to-image Models.** Early works on text-to-image synthesis based on GANs [53, 57, 21, 50] were limited to small-scale and object-centric datasets, due to training difficulties of GANs. Auto-

regressive models pioneered the use of large-scale data for text-to-image generation [33, 34, 41, 8, 52]. However, they typically suffer from high computation costs and error accumulation. An emerging trend is large-scale text-to-image diffusion models, which are the current state-of-the-art in image synthesis, offering unprecedented image fidelity and language reasoning capabilities. Research efforts have focused on improving their image quality, controllability, and expanding the type of conditional inputs [22, 9, 2, 54, 37, 38, 8, 29, 40, 51]. In the realm of image editing, diffusion models are now at the forefront, providing rich editing capabilities through text descriptions and conditional inputs. In this work, we investigate how to use visual prompts to guide image edits with diffusion models.

**Image Editing.** In the beginning, image editing was primarily done within the image space. Previous GAN-based approaches utilized the meaningful latent space of GANs to perform editing [18, 1, 24, 11]. The inversion technique [36, 7, 42, 35] has been used to obtain the latent features of the input image, perform editing in the latent space, and revert the output to the image space. More recently, with the support of CLIP [32], a bridge between images and texts, image editing can now be guided by text prompts [31, 11]. Recent models for text-conditioned image editing have leveraged CLIP embedding guidance and text-to-image diffusion models to achieve state-of-the-art results for a variety of edits. There are three main directions for research in this area: (1) zero-shot (exploiting the CLIP embedding directions, stochastic differential equations, or attention control) [13, 30, 26, 27, 20]; (2) optimizing text prompts and/or diffusion models [19, 39, 10, 16, 56]; and (3) fine-tuning diffusion models on a supervised dataset [4, 54]. In contrast to prior works that rely on text prompts to guide the editing, we aim to leverage visual prompts to better assist the process.

**Prompt Tuning.** Diffusion models have shown stirring results in text-to-image generation, but they can struggle to comprehend specific or novel concepts. Several works have focused on addressing this issue. Textual Inversion [10] learns a specialized token for new objects, which can later be plugged in with natural language to generate novel scenes. ReVersion [16] learns a specified text prompt for relation properties between two sets of images. Although a continuous prompt can be more task-specific, a discrete prompt is typically easier to manipulate by users. PEZ [48] proposes a method to discover prompts that can retrieve similar concepts of given input images. Instead of learning novel concepts for image generation, our work focuses on learning the *transformation* between an example pair of images that is better suited for image editing.

**Visual Prompting.** Since proposed in NLP [5], prompting has been adapted by computer vision researchers. Unlike traditional methods that require separate models for each downstream task, visual prompting utilizes in-context learning to solve different tasks during inference. The first application of visual prompts was proposed by [3], where an example and query image are combined to form a grid-image. The task solver fills in the missing portion, which contains the answer. They showed that the task solvers can perform effectively on several tasks with only training on a dataset of Computer Vision figures. Later, [44] and [45] expanded the framework to increase the number of tasks that can be solved. Recently, Prompt Diffusion [47] introduced a diffusion-based foundation for in-context learning. Although it shows high-quality in-context generation, a text prompt is still needed. Similar to textual prompts, not all visual prompts perform equally well. There are ongoing efforts to understand how to design a good example pair [55]. The most similar idea to visual prompting is the seminal work of Image Analogies [14] and its variants [43, 23]. However, these works rely on pixel-to-pixel correspondences between example and test images, and struggle to transfer high-level idea edits (e.g., "add a dog" or "turn it into a monkey"). Despite the success of visual prompting in solving a wide range of standard computer vision tasks, the question of whether one can use visual prompting for image editing remains unanswered.

## 3 Framework

In this section, we present our approach for enabling image editing via visual prompting. First, we provide a brief background on text-conditioned image editing diffusion models (Section 3.1). Section 3.2 and 3.3 describe how to invert visual prompts into text-based instructions. Finally, our full Visual Instruction Inversion algorithm is given in Section 3.4.

Let $\{x, y\}$ denote the before-and-after example of an edit. Our goal is to learn a text-based edit $c_T$ that captures the editing direction from $x$ to $y$. Once learned, $c_T$ can be applied to any new input image $x'$, to obtain an edited image $y'$ that undergoes a similar transformation: $x \to y \approx x' \to y'$.

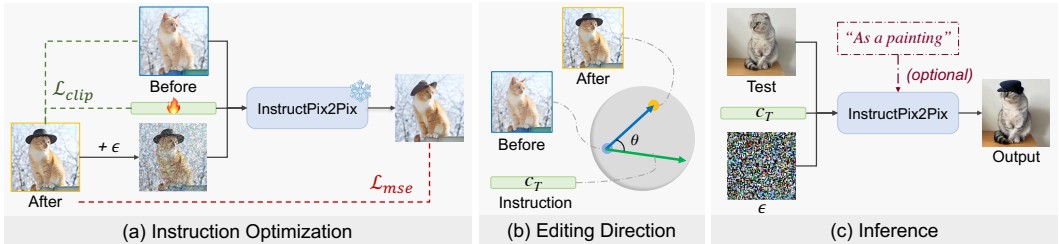

Figure 3: **Our framework**. (a) Given an example before-and-after image pair, we optimize the latent text instruction that converts the "before" image to the "after" image using a frozen image editing diffusion model. (b) We leverage the CLIP embedding space to help learn the editing direction. (c) Once learned, the instruction can be applied to a new image to achieve the same edit. Optionally, the user can also combine the learned instruction with a natural text prompt to create a hybrid instruction.

To avoid confusion, we use only one image pair example to describe our approach. However, it is worth noting that our algorithm still holds for an arbitrary number of example pairs.

### 3.1 Preliminaries

Diffusion models for image generation are trained on a sequence of gradually noisier image $x$ over a series of timesteps $t = 1, \dots, T$. The goal is to learn a denoising autoencoder $\epsilon_\theta$, which predicts a denoised variant of a noisy version of $x$ at each timestep $t$, commonly denoted as $x_t$ [15]. Initially, this approach was implemented in pixel-space [6], but it has now been extended to the latent space for faster inference and improved quality [37]. Here, prior to the diffusion process, image $x$ is encoded by an encoder $\mathcal{E}$ to obtain the latent image $z_x$, which is subsequently decoded by a decoder $\mathcal{D}$ to convert the latent image back to the image space. The objective function is defined as follows:

$$\mathcal{L} = \mathbb{E}_{\mathcal{E}(x), \epsilon \sim \mathcal{N}(0,1), t} \|\epsilon - \epsilon_\theta(z_{x_t}, t)\|_2$$

To enable diffusion models to take text prompts as conditional inputs, [37] introduced a domain-specific encoder $\tau_\theta$ that projects text prompts to an intermediate representation $c_T$. This representation can then be inserted into the layers of the denoising network via cross-attention:

$$\mathcal{L} = \mathbb{E}_{\mathcal{E}(x), c_T, \epsilon \sim \mathcal{N}(0,1), t} \|\epsilon - \epsilon_\theta(z_{x_t}, t, c_T)\|_2$$

Conditioned on a text description $c_T$, diffusion models can synthesis stunning images. However, they are still not completely well-suited for image editing. Suppose that we want to edit image $x$ to image $y$, conditioned on text prompt $c_T$. Text prompt $c_T$ then needs to align with our desired edit $x \rightarrow y$ and fully capture the visual aspects of $x$, which can be difficult. There are methods to help discover text prompts that can retrieve similar content [48, 28], however, they typically cannot accurately describe all aspects of the input image $x$. To address this challenge, the idea of adding the input image to the denoising network was proposed in [4, 40]. The input image $x$ can then be encoded as $c_I = \mathcal{E}(x)$ and concatenated to the latent image $z_{y_t}$, jointly guiding the editing process with text prompt $c_T$. Based on this idea, InstructPix2Pix [4] fine-tunes the text-to-image diffusion model in a supervised way to perform image editing. Its objective function is changed accordingly, as it now learns to denoise a noisy version of $y$, which is an edited image of $x$ based on editing direction $c_T$:

$$\mathcal{L} = \mathbb{E}_{\mathcal{E}(y), c_T, c_I, \epsilon \sim \mathcal{N}(0,1), t} \|\epsilon - \epsilon_\theta(z_{y_t}, t, c_T, c_I)\|_2 \tag{1}$$

### 3.2 Learning to Reconstruct Images

Prior textual inversion methods [10, 39, 16] all utilize an image reconstruction loss. However, their aim is to learn to capture the essence of the concept in the image so that it can be synthesized in new contexts, but not to faithfully follow pixel-level details of the input image which are required for image editing. The closest idea to ours is [19], but it needs to fine-tune the diffusion model again for each edit and input image. We instead exploit a pre-trained text-conditioned image editing model, which offers editing capabilities, while avoiding additional fine-tuning.

Given only two images $\{x, y\}$ which represent the "before" and "after" images of an edit $c_T$, the first and foremost objective is to recover image $y$. We employ a pretrained text-conditioned image editing model proposed in [4], where we optimize the instruction $c_T$ based on the supervised pair $\{x, y\}$. In

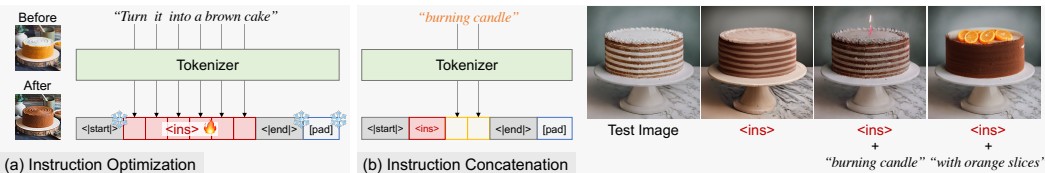

Figure 4: **Instruction details**. (a) Instruction Optimization: We only optimize a part of the instruction embedding $c_T$, called . (b) Hybrid instruction: During test time, we can add extra information into the learned instruction $c_T$ to further guide the edit.

our case, conditional image $c_I$ is the "before" image $x$, and target image is the "after" image $y$. The objective function is then adopted from Eq. 1 as:

$$\mathcal{L}_{mse} = \mathbb{E}_{\mathcal{E}(y),c_T,z_x,\epsilon \sim \mathcal{N}(0,1),t} \|\epsilon - \epsilon_\theta(z_{y_t}, t, c_T, z_x)\|_2 \qquad (2)$$

### 3.3 Learning to Perform Image Editing

If we rely only on the image reconstruction constraint (Eq. 2), we may learn a description of the edited image $y$, instead of the desired editing instruction. [31, 11, 30] has shown that the CLIP embedding [32] is a good indicator of the editing direction. [30] uses GPT-3 [5] to generate a set of sentences for the "before" and "after" domains of an edit; for example, cat $\leftrightarrow$ dog. The mean difference between the CLIP embeddings of these sentences represents the text editing direction "before" $\leftrightarrow$ "after".

In our case, we can use the difference between the CLIP embeddings of the "after" and "before" images to help learn the edit. Specifically, for an example pair $\{x, y\}$, we compute the image editing direction $\Delta_{x \rightarrow y}$ as:

$$\Delta_{x \rightarrow y} = \mathcal{E}_{\text{clip}}(y) - \mathcal{E}_{\text{clip}}(x)$$

We encourage the learned instruction $c_T$ to be aligned with this editing direction (Figure 3b). To this end, we minimize the cosine distance between them in the CLIP embedding space:

$$\mathcal{L}_{clip} = \text{cosine}(\Delta_{x \rightarrow y}, c_T) \qquad (3)$$

### 3.4 Image Editing via Visual Prompting

Finally, given an example before-and-after image pair $\{x, y\}$, we formulate the visual prompting as an instruction optimization using our two constraints: Image reconstruction loss (Eq. 2) and CLIP loss (Eq. 3). We provide an illustration of our framework in training and testing in Figure 3a,c, and pseudocode in Algorithm 1. Our algorithm also holds for $n$ example pairs $\{(x_1, y_1), \ldots (x_n, y_n)\}$. In this case, $\Delta_{x \rightarrow y}$ becomes the mean difference of all examples, and at each optimization step, we randomly sample one pair $\{x_i, y_i\}$.

Once $c_T$ is learned, we can apply it to a new image $x_{test}$ to edit it into $y_{test}$. Moreover, our designed approach allows users to input extra information, enabling them to combine the learned instruction $c_T$ with an additional text prompt (Figure 3c). To that end, we optimize a fixed number of tokens of $c_T$ only, which provides us with the flexibility to concatenate additional information to the learned instruction during inference (Figure 4b). This allows us to achieve more fine-grained control over the resulting images, and is the final default approach.

## 4 Evaluation

We compare our approach against both image-editing and visual prompting frameworks, on both synthetic and real images. In Section 4.2, we present qualitative results, followed by a quantitative comparison in Section 4.3. Both quantitative and qualitative results demonstrate that our approach not only achieves competitive performance to state-of-the-art models, but also has additional merits in specific cases. Additional qualitative results can be found in the Appendix.

**Algorithm 1** **Vis**ual Instruction Inversion (**VISII**)

1: **Input**: An example pair $\{x, y\}$
2:     Pretrained denoising model $\epsilon_\theta$; Image encoder $\mathcal{E}$; CLIP encoder $\mathcal{E}_{clip}$
3:     Number of optimization steps $N$; Number of timesteps $T$
4:     Hyperparameters $\lambda_{clip}$, $\lambda_{mse}$; Learning rate $\gamma$
5:     // Start optimization
6:     Initialize $c_T$                                                          ▷ Initialize instruction
7:     Encode $z_x = \mathcal{E}(x)$;    $z_y = \mathcal{E}(y)$                            ▷ Encode image
8:     Compute $\Delta_{x \to y} = \mathcal{E}_{\text{clip}}(y) - \mathcal{E}_{\text{clip}}(x)$          ▷ Compute editing direction
9: **for** $i = 1, \cdots, N$ **do**
10:     Sample $t \sim \mathcal{U}(0, T)$; $\epsilon \sim \mathcal{N}(0, 1)$           ▷ Sample timestep and noise
11:     $z_{y_t} \leftarrow$ add $\epsilon$ to $z_y$ at timestep $t$     ▷ Prepare noisy version of $z_y$ at timestep $t$
12:     $\hat{\epsilon} = \epsilon_\theta(z_{y_t}, t, c_T, z_x)$                     ▷ Predict noise condition on $x$
13:     $\mathcal{L} = \lambda_{mse}\|\epsilon - \hat{\epsilon}\|_2 + \lambda_{clip}(\text{cosine}(c_T, \Delta_{x \to y}))$     ▷ Compute losses
14:     Update $c_T = c_T - \gamma \nabla \mathcal{L}$
15: **end for**
16: **Output**: $c_T$

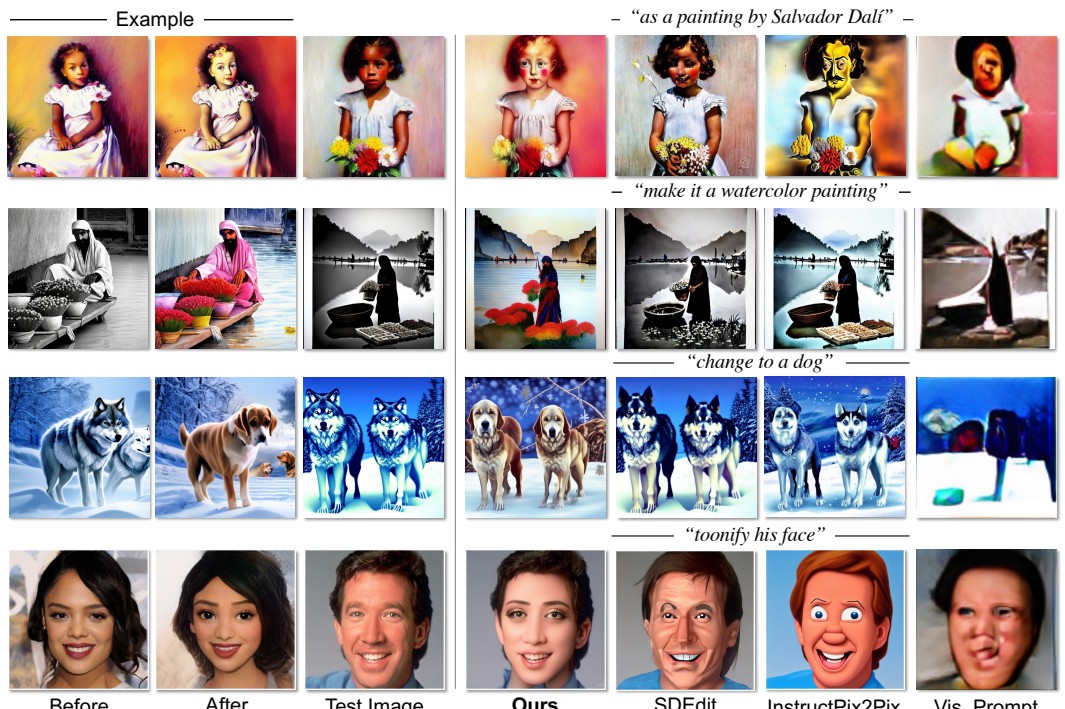

Figure 5: **Qualitative comparisons.** Our method learns edits from example pairs and thus can produce visually closer edited images to the target example than other state-of-the-art baselines.

## 4.1 Experimental Settings

**Training Setting.** We use the frozen pretrained InstructPix2Pix [4] to optimize the instruction $c_T$ for $N = 1000$ steps, $T = 1000$ timesteps. We use AdamW optimizer [25] with learning rate $\gamma = 0.001$, $\lambda_{mse} = 4$, and $\lambda_{clip} = 0.1$. Text guidance and image guidance scores are set at their default value of 7.5 and 1.5, respectively. All experiments are conducted on a $4 \times$ NVIDIA RTX 3090 machine.

**Dataset.** We randomly sampled images from the Clean-InstructPix2Pix dataset [4], which consists of synthetic paired before-after images with corresponding descriptions. In addition, we download paired photos from [17] to test the models. Since some real images do not have edited versions, we utilize [54] with manual text prompts to generate the after images with different edits.

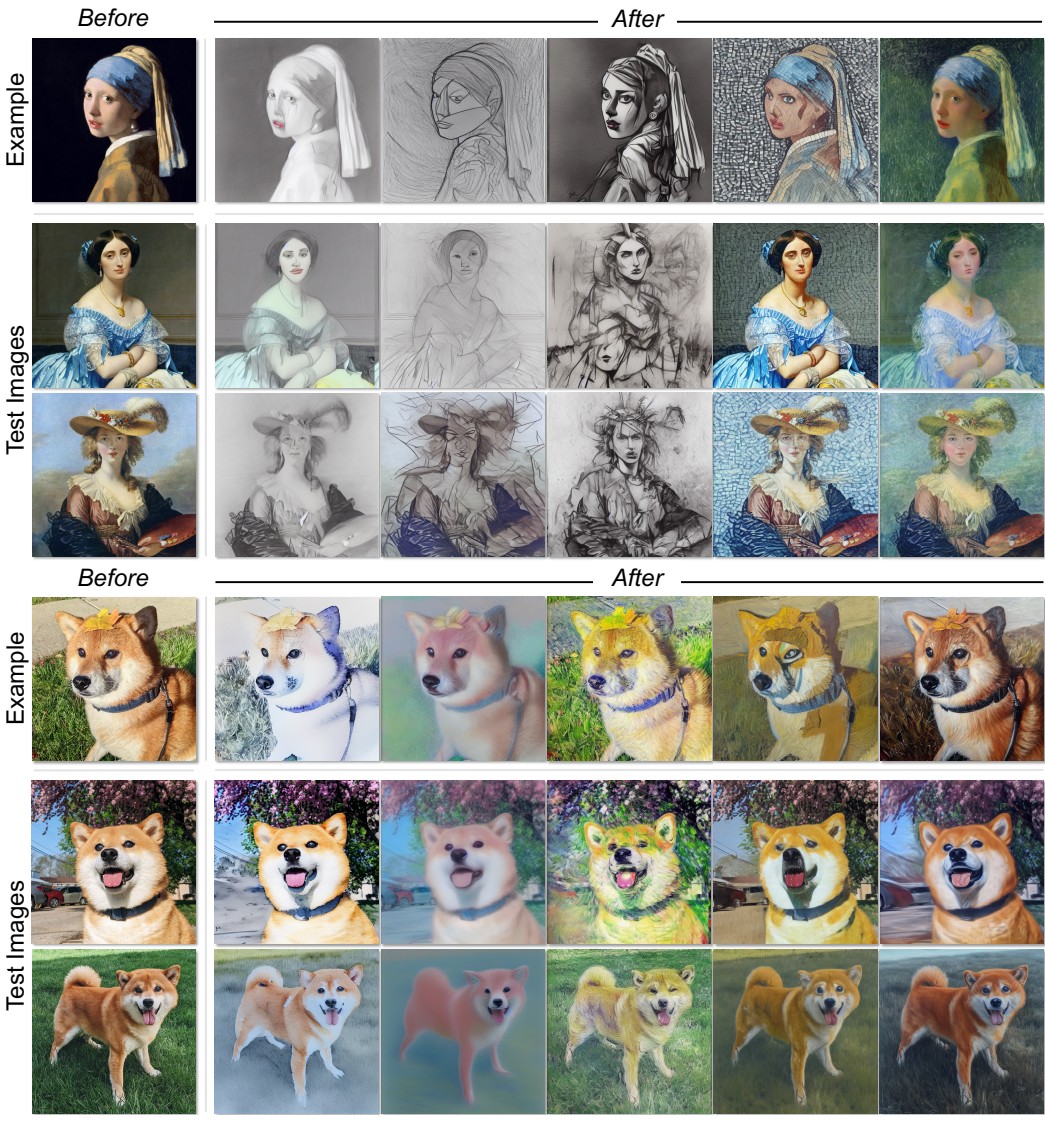

Figure 6: A variety of edits can be performed for "Turn it into a drawing/ painting" (Zoom in for details).

**Evaluation Metrics.** Following [4], we assess the effectiveness of our approach using the Directional CLIP similarity [11] and Image CLIP similarity metrics. However, as the CLIP directional metric does not reflect the transformation similarity between the before-after example and before-after output pair, we propose an additional metric called the Visual CLIP similarity. Specifically, we compute the cosine similarity between the before-after example pair and the before-after test pair as follows: $s_{visual} = 1 - \text{cosine}(\Delta_{x \to y}, \Delta_{x' \to y'})$.

**Baseline Models.** We compare our approach to two main categories of baselines: Image Editing and Visual Prompting. For image editing, we compare against InstructPix2Pix [4] and SDEdit [26], which are the state-of-the-art. We directly use the ground-truth editing instruction for InstructPix2Pix and after descriptions for SDEdit. For real images, we manually write instructions and descriptions for them, respectively. For visual prompting, we compare our approach against Visual Prompting [3]. The Visual Prompting code is from the author's official repository, while SDEdit and InstructPix2Pix codes are from HuggingFace.

## 4.2 Qualitative Results

Figure 5 presents qualitative comparisons. As can be seen, Visual Prompting [3] fails to perform the image editing task. Text-conditioned image editing frameworks, InstructPix2Pix [4] and SDEdit [26],

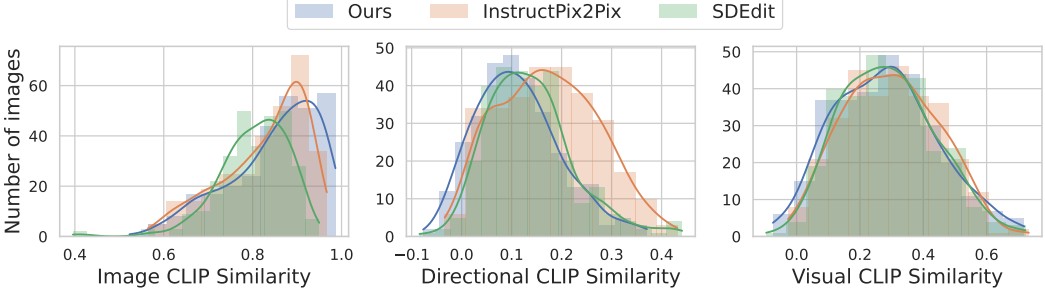

Figure 7: **Quantitative comparison.** Histogram of Image, Directional, and Visual CLIP similarity scores. Our results are comparable to state-of-the-art text-conditioned image editing frameworks.

can edit images based on the provided text prompts, but can fall short in producing edited images that are visually close to the "after" example. In contrast, our approach can learn the edit from the given example pair and apply it to new test images. For example, in wolf ↔ dog (Fig. 5, row 3), we not only achieve successful domain translation from wolf to dog, but we also preserve the color of the dog's coat. Please refer to the Appendix for more qualitative results.

Figure 6 demonstrates the advantage of visual prompting compared to text-based instruction. We can see that one text instruction can be unclear to describe specific edits. For example, the instruction "Turn it into a drawing" or "Make it a painting" can have multiple interpretations in terms of style and genres. In these cases, by showing an example pair, our method can learn and replicate the distinctive characteristics of each specific art style.

### 4.3 Quantitative Results

We perform quantitative evaluation of our method against two baselines: InstructPix2Pix [4] and SDEdit [26]. Since Visual Prompting was not effective in performing image editing tasks, we did not include it in our comparison. We randomly sampled 300 editing directions, resulting in a total of 1030 image pairs, from the Clean-InstructPix2Pix dataset.

We analyze the histograms of Image, Directional, and Visual CLIP Similarity (Figure 7). Results indicate that our method performs competitively to the baselines. In terms of Directional CLIP Similarity, InstructPix2Pix achieves the highest score, as it can make large changes the input image toward the editing instruction. Our method scores similarly to SDEdit, indicating that our approach can also perform well in learning the editing direction. Our approach is the most faithful to the input image, as reflected by the highest Image CLIP Similarity scores. Finally, for Visual CLIP Similarity, which measures the agreement between the changes in before-after example and before-after test images, our approach performs nearly identically to the two state-of-the-art models.

## 5  Analysis

We next conduct an in-depth study to better understand our method. For all of the studies below, we sample 100 editing directions (resulting in 400 before-and-after pairs in total) from the Clean-InstructPix2Pix [4] dataset. We show that both the CLIP loss and instruction initialization are critical for achieving optimal performance. Additionally, we present some interesting findings regarding the effects of random noise, which can lead to variations in the output images.

**Losses.** We ablate the effect of each loss in Table 1. The additional CLIP loss helps improve scores in Visual and Directional CLIP Similarity [11], which reflect the editing directions. This shows that the CLIP loss encourages the learned instruction to be aligned with the target edit.

**Initialization.** Prior work utilizes a coarse user text prompt (e.g., "sculpture" or "a sitting dog") for textual initialization [10, 19], which can be practical, but may not always be effective. The reason is that natural text prompts can be misaligned with the model's preferred prompts [12]. We can also optimize upon a user's coarse input, however, we find that it is more effective to initialize the instruction vector $c_T$ to be somewhere close to the editing target; i.e., a caption of the "after" image.

Table 1: **Quantitative Analysis.** We report Image, Directional, and Visual CLIP Similarity scores. Despite learning from only one example pair, our approach performs competitively to state-of-the-art image editing models. ("Direct.": "Directional"; #: number of training pairs; "Init": Initialization of instruction; "GT": Ground-truth instruction; "Cap.": Image captioning of "after" image.)

| | Losses | | Init. | | Random noise | | | Fixed noise | | |
|---|---|---|---|---|---|---|---|---|---|---|
| # | MSE | CLIP | GT | Cap. | Img ↑ | Direct. ↑ | Visual ↑ | Img ↑ | Direct. ↑ | Visual ↑ |
| | *ground-truth* | | | | 0.824 | **0.196** | **0.301** | - | - | - |
| | *no training* | | | ✓ | **0.866** | 0.090 | 0.199 | - | - | - |
| 1 | ✓ | | ✓ | | 0.841 | 0.120 | 0.247 | 0.854 | 0.105 | 0.223 |
| 1 | ✓ | | | ✓ | 0.845 | 0.115 | 0.254 | 0.861 | 0.110 | 0.225 |
| 1 | ✓ | ✓ | ✓ | | 0.838 | 0.131 | 0.231 | 0.852 | 0.102 | 0.236 |
| 1 | ✓ | ✓ | | ✓ | 0.823 | 0.126 | 0.299 | 0.847 | 0.113 | 0.251 |
| 1 | ✓ | ✓ | | ✓ | 0.823 | 0.126 | 0.299 | 0.847 | 0.113 | 0.251 |
| 2 | ✓ | ✓ | | ✓ | 0.791 | 0.141 | 0.292 | 0.826 | 0.117 | 0.253 |
| 3 | ✓ | ✓ | | ✓ | 0.780 | 0.148 | 0.283 | 0.805 | 0.132 | 0.256 |
| 4 | ✓ | ✓ | | ✓ | 0.798 | 0.148 | 0.280 | 0.812 | 0.133 | 0.260 |

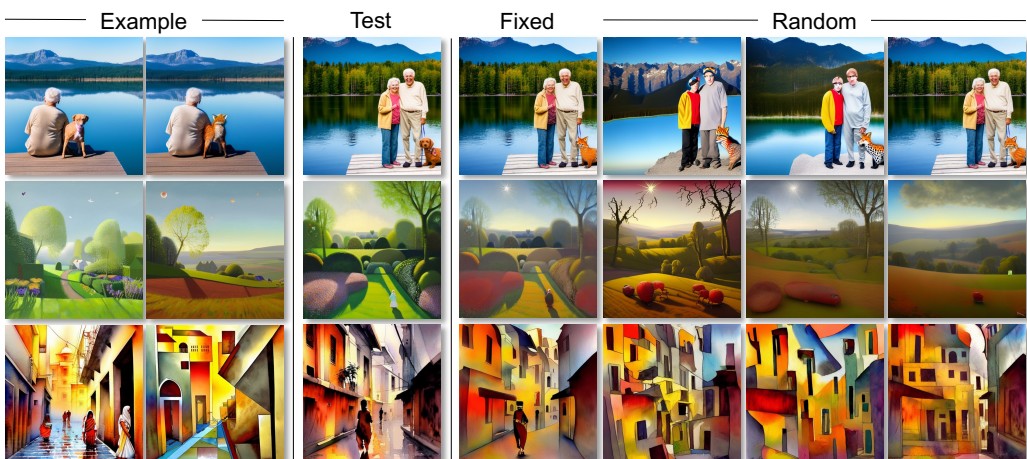

Figure 8: **Fixed noise leads to more balanced results**. Different noises can lead to large variations in the output. Using the same training noises yields a balanced trade-off between editing manipulation and image reconstruction.

We evaluate both initialization strategies, including user input and captioning. To mimic a user's coarse input, we directly use ground-truth editing instructions from Clean-InstructPix2Pix dataset [4]. We employ [48] to generate captions for "after" images. Results are shown in Table 1. As expected, directly using the caption as the instruction to InstructPix2Pix will not yield good results (Row 2), but initializing our model's learned instruction from the caption helps to improve the Visual and Image CLIP Similarity scores. This indicates that the learned instruction is more faithful to the input test image that we want to edit, while still retaining editing capabilities.

**Noises.** Text-conditioned models generate multiple variations of output images depending on the sampled noise sequence. This is true for our approach too. However, for image editing, we would prefer outputs that best reflect the edit provided in the before-and-after image pair, and preserve the test image as much as possible apart from that edit. We find that using identical noises from training in test time can help achieve this. Specifically, denote the noises sampled during the training optimization timesteps $t = 1 \ldots T$, as $\{\epsilon_1, \ldots \epsilon_T\}$, which are added to the latent image $z_y$. We reuse the corresponding noises in the backward process during test time to denoise the output images. This technique helps to retain the input image content, as shown in Table 1. However, there is a trade-off between aggressively moving toward the edit, and retaining the conditional input. It is worth noting that in test time, we can also use random noises for denoising if desired. We visualize this phenomenon in Figure 8.

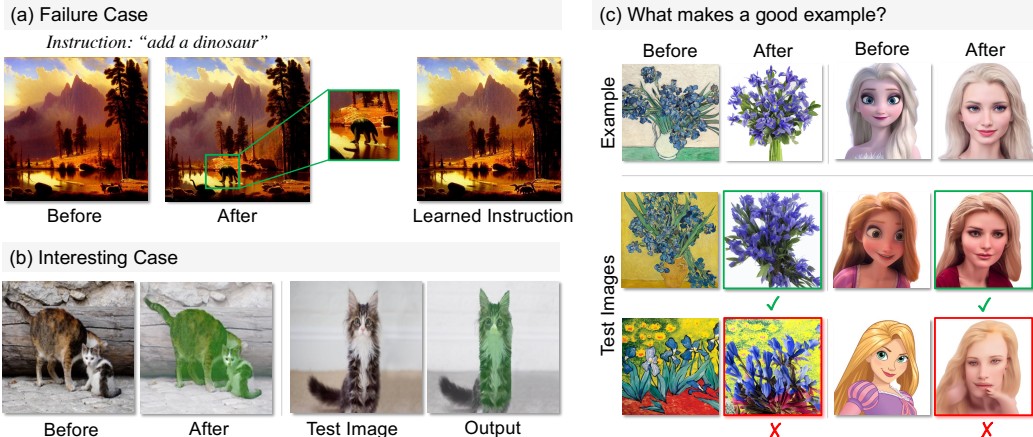

Figure 9: **Discussion.** (a) Failure Case: Our model can fail to capture fine details. (b) Interesting case: By preparing an image and segmentation pair, we can perform image segmentation. (c) Quality of example pair: One example does not work equally well for different test images.

**Hybrid instruction.** Finally, our approach allows users to incorporate additional information into the learned instruction. Specifically, we create a hybrid instruction by concatenating the learned instruction with the user's text prompt. This hybrid instruction better aligns with the given example pair while still following the user's direction. In Figure 1, we transform "cat" → "watercolor cat". We demonstrate how concatenating extra information to the learned instruction () enables both image editing (changing to a watercolor style) and domain translation (e.g., "cat" → "tiger"). The painting style is consistent with the before-and-after images, while the domain translation corresponds to the additional information provided by the user. Figure 12 provides more qualitative examples. Applying InstructPix2Pix [4] often does not yield satisfactory results, as the painting style differs from the reference image.

## 6  Discussion and Conclusion

We presented a novel framework for image editing via visual prompt inversion. With just one example representing the "before" and "after" states of an image editing task, our approach achieves competitive results to state-of-the-art text-conditioned image editing models. However, there are still several limitations and open questions left for future research.

One major limitation is our reliance on a pre-trained model, InstructPix2Pix. As a result, it restricts our ability to perform editing in the full scope of diffusion models, and we might also inherit unwanted biases. Additionally, there are cases where our model fails, as shown in Figure 9a, where we fail to learn "add a dinosaur" to the input image, presumably because it is very small.

As we address the question of effectively using visual prompting with diffusion models, one might ask an interesting question in the reverse direction: Can diffusion models be used as a task solver for downstream computer vision tasks? We find out that by preparing a foreground segmentation as a green area, we can learn instructions and apply them to new images to obtain corresponding segmentations (Figure 9b). However, further research is needed to fully explore this question. We acknowledge that this question is beyond the scope of our current study, which is primarily focused on image editing. Additionally, visual in-context learning has been shown to be sensitive to prompt selection [55, 3]. Figure 9c shows cases where one example may not fit all test images. This shows that there are open questions regarding what makes a good example for image editing.

**Acknowledgements.** This work was supported in part by NSF CAREER IIS2150012, Adobe Data Science award, Sony Focused Research award, and Institute of Information & communications Technology Planning & Evaluation (IITP) grant funded by the Korea government (MSIT) (No. RS-2022-00187238, Development of Large Korean Language Model Technology for Efficient Pre-training).

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

# Appendix

## A   Textual Inversion vs. Visual Instruction Inversion

Textual Inversion [10, 39] is a method to invert a visual concept into a corresponding representation in the language space. In particular, given (i) a text-to-image pre-trained model and (ii) some images describing a visual concept (e.g., a particular kind of toy; Figure 10 bottom row), Textual Inversion learns new "words" in the embedding space of the text-to-image model to represent those visual concepts. Once these "words" are learned for that concept, they can be plugged into arbitrary textual descriptions, just like other English words, which can then be used to create the target visual concept in different contexts. Instead of learning the representation for an isolated visual concept, our approach (Visual Instruction Inversion), learns the *transformation* from a before-and-after image pair. This learned transformation is then applied to a test image to achieve similar edit "before" → "after".

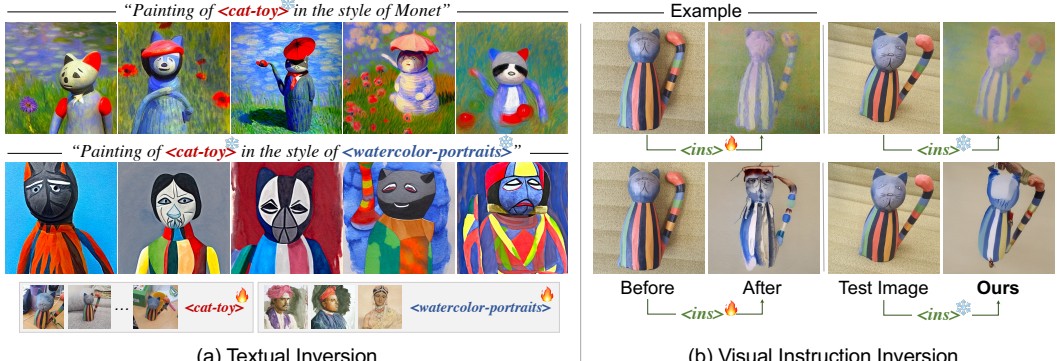

Figure 10: **Ours vs. Textual Inversion**. (a) Textual Inversion inverts a visual concept or object (e.g., a particular `<cat-toy>`) into a word embedding. This optimized word embedding can then be combined with a textual description to generate novel scenes. (b) Our Visual Instruction Inversion learns the transformation ``: "before" → "after" in a given before-and-after image pair. This learned instruction can then be applied to new test images to perform the same edit.

**Applicability of Textual Inversion for image editing.** Given these differences with our proposed method, we now try to see if Textual Inversion can be used for image editing. Textual Inversion can generate a "painting of a `<cat-toy>` in the style of Monet" by using the learned word `<cat-toy>` from example images (Figure 10a, Row 1). However, the synthesized images often only capture the essence of the objects, and disregard the details of the input images. As a result, textual inversion is suitable for novel scene composition, but is not effective for image editing.

On the other hand, our Visual Instruction Inversion does not learn novel token representations for objects or concepts. Instead, we learn the edit instruction from before-and-after pairs, which can be applied to any test image to obtain corresponding edits. This allows us to achieve fine-grained control over the resulting images. For example, by providing a photo of `<cat-toy>`, one before and one in a specific impressionist style, we learn the transformation from before to impressionist, denoted as ``. Once learned, this instruction can be applied to new `<cat-toy>` images to achieve the same impressionist painting style, without losing the fine details of the test image (Figure 10b, Row 1).

One might suggest an alternative approach to image editing using Textual Inversion, which involves learning two tokens: one for the object and another for the style (e.g., "Painting of `<cat-toy>` in the style of `<watercolor-portraits>`"). Figure 10a (Row 2) shows the results of this approach. As can be seen, Textual Inversion still often introduces significant changes that deviate from the original input image. Thus, Textual Inversion is not suitable for accurate image editing.

# B   Additional Qualitative Comparisons

We present qualitative comparisons with other state-of-the-art text-conditioned image editing methods, Imagic [19] and Null-text Inversion [27] (Figure 11). These methods can generate outputs based on given text prompts, such as "A watercolor painting of a cat" (Row 3). However, the outputs often do not match the given reference. The text prompts can also be ambiguous and result in unsatisfactory outputs, as illustrated by the case of "A character in a Pixar movie" (Row 1).

Another challenge is the inconsistency of text-conditioned models, where the same text prompt can produce different outputs for different test images. For example, the text prompt "A frozen waterfall" (Row 6) generates different water colors (blue vs. white) when applied to different test images (Before-and-after pair is from [27]). Our method is more consistent in this case, as the learned instruction might have learned the water color.

# C   Implementation Details

We use the pretrained `clip-vit-large-patch14` as the CLIP Encoder in our approach. For instruction initialization [48], we set the caption length for after image to 10 tokens. However, this specific caption length does not affect the optimization algorithm. We can optimize initialization

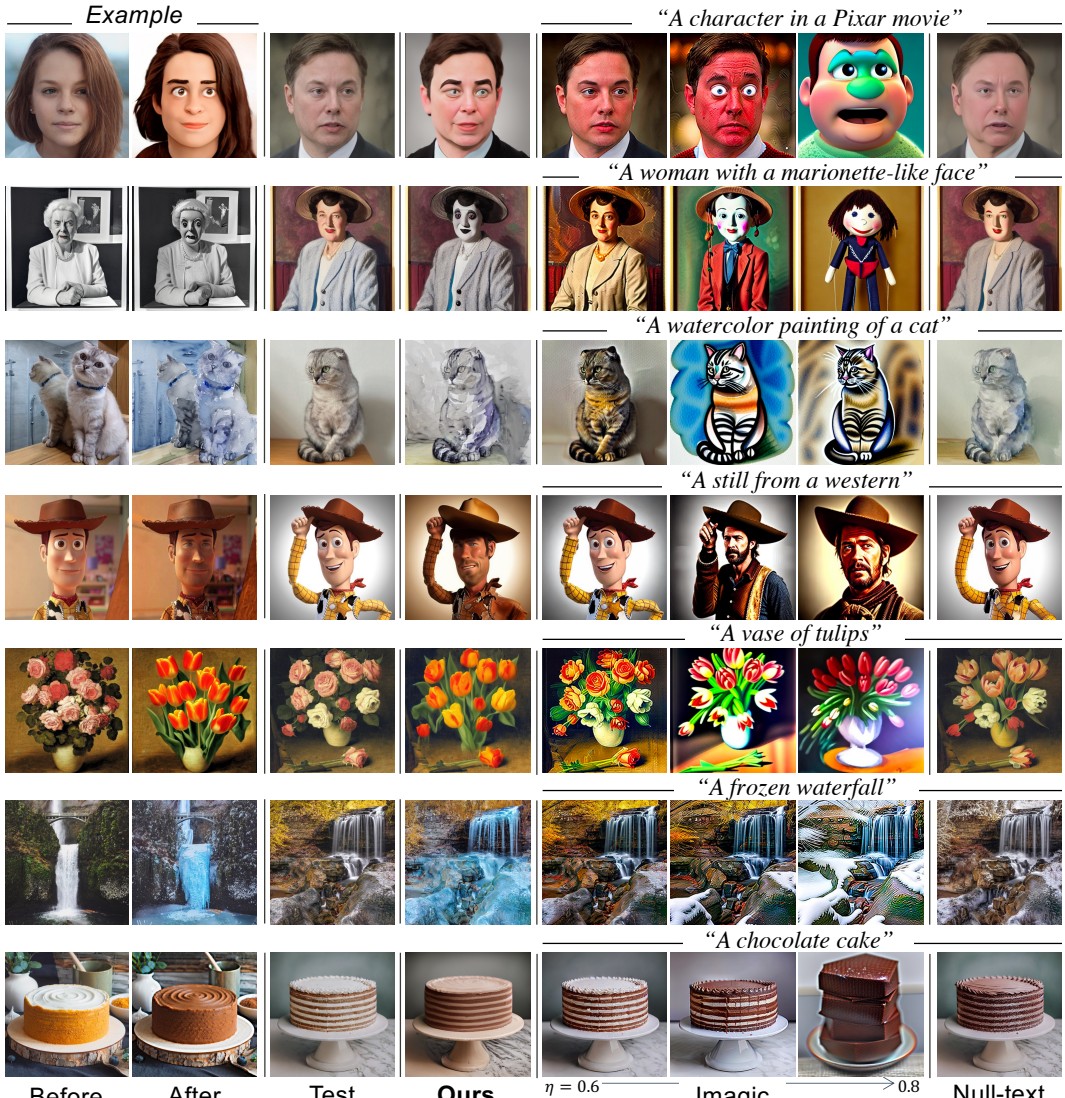

Figure 11: **Additional qualitative comparisons** to Imagic [19] and Null-text Inversion [27]. Imagic and Null-text Inversion fail to match the reference image as they perform edits based on ambiguous text prompts (Row 1-4); or exhibit inconsistency in producing outputs for the same prompt across test images (Row 6). In contrast, our method produces visually closer edited images to the before-after pair while demonstrating improved consistency by using the learned instructions.

instructions of varying lengths (up to 77 tokens). It takes roughly 7 minutes to optimize for one edit, and 4 seconds to apply the learned instruction to new images.

Specifically, during the optimization process, we freeze the tokens representing the start of text (`<|startoftext|>`), end of text (`<|endoftext|>`), and all padding tokens after end of text (`<|endoftext|>`). We only update the tokens inside the text prompt, called `` (between `<|startoftext|>` and `<|endoftext|>`) (Figure 4a).

## Photo Attribution

- Elsa (Human): reddit.com/r/Frozen
- Disney characters: princess.disney.com
- Toy Story characters: toystory.disney.com
- Toonify faces: toonify.photos

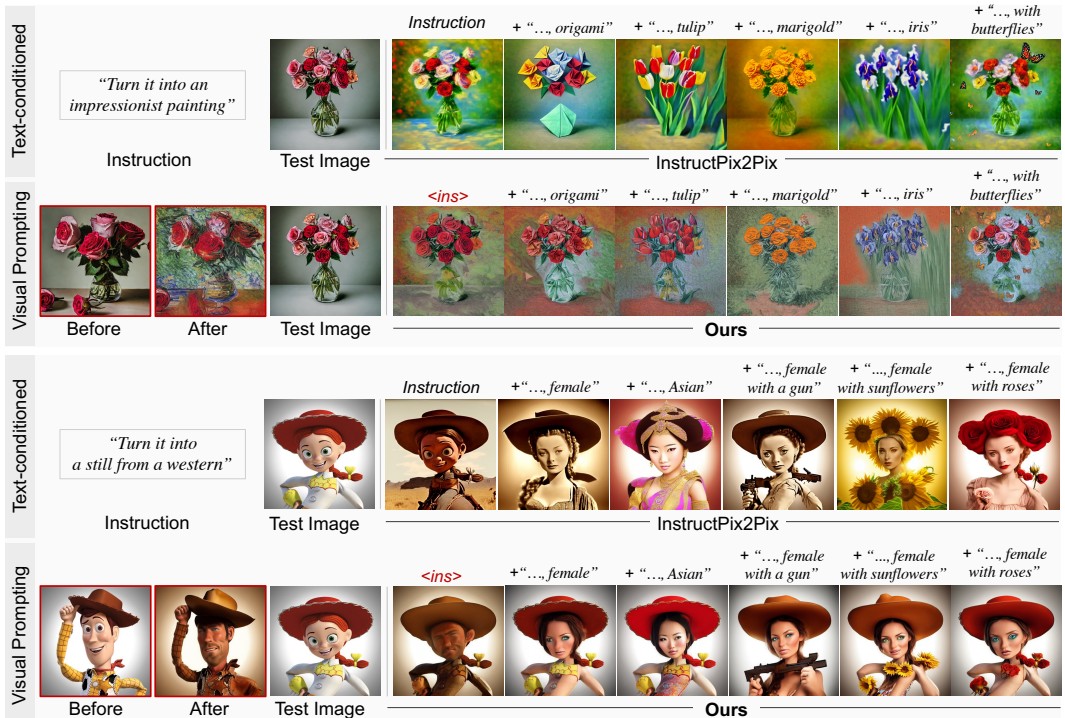

Figure 12: **Hybrid instruction**. We can concatenate extra information into the learned instruction $c_T$ to navigate the edit. (Zoom in for details.)

- Girl with a Pearl Earring: wikipedia/girl-with-a-pearl-earring
- Mona Lisa: wikipedia/mona-lisa
- The Princesse de Broglie: wikipedia/Princesse-de-Broglie
- Self-portrait in a Straw Hat: wikipedia/self-portrait-in-a-straw-hat
- Bo the Shiba and Mam the Cat: instagram/avoshibe
- `<cat-toy>` and `<watercolor-portraits>` concept: huggingface.co/sd-concepts-library
- Gnochi cat, waterfall, and cake images are from Imagic [19] and Null-text Inversion [27].

