# OpenReview forum: "Visual Instruction Inversion: Image Editing via Image Prompting"
_NeurIPS.cc/2023/Conference — NeurIPS 2023 poster_

### Official Review · Reviewer_G8X8 · 2023-06-25

**Soundness:** 3 good
**Presentation:** 4 excellent
**Contribution:** 3 good
**Rating:** 7
**Confidence:** 4

**Summary:**

The authors propose a method for finding a text-based editing direction extracted from a pair of “before” and “after” images depicting the desired edit. Using a fixed, pretrained diffusion model (in this case Stable Diffusion), the authors optimize the text-conditioning embedding to align with the CLIP-space direction between the two images. This is done will minimizing a reconstruction loss to ensure that the main details of the original image are preserved. Qualitative and quantitative results show the effectiveness of the method when compared to existing approaches (InstructPix2Pix, SDEdit, and other image editing techniques such as Imagic). The authors also provide a useful analysis of how the noise and initialization influence the generation process when aiming to edit a given image.

**Strengths:**

- The authors present a simple, yet effective technique for extracting text-based editing directions in a pretrained diffusion model. This is particularly useful when a desired edit is difficult to describe with language.
- Extracting this direction requires only a single exemplar pair, making it applicable and easy to use across many settings.
- The instruction concatenation technique is very interesting and beneficial for providing users more control in cases where the extract direction is not perfect. I almost missed this, but found it to be a very interesting addition. I would consider adding this to the main paper if possible as it provides an additional advantage over alternative techniques.
- Comparisons are performed to a variety of techniques including InstructPix2Pix and SDEdit. Further evaluations are performed on Textual Inversion, Imagic, and Null-Text Inversion. The results provided by the authors seem to out-perform existing techniques across a variety of edits and images.

**Weaknesses:**

- It appears that the proposed method is more effective on style-based edits, as I could not find any examples to require a large change in the structure of the object. How does the method fair when the prompt pair depicts a change in structure (e.g., sitting/jumping)? I could not find examples or a discussion on this.
- In Figure 7 the authors show that a single example may not generalize well to all different test sets. This is expected, but I am wondering how sensitive the method is to the choice of the exemplar pair. Assessing this could be challenging, but one possible way to assess this could be evaluating the success rate of a single input pair across a set of test images. This could be done either quantitatively (e.g., using CLIP-space similarities) or qualitatively using a user study, if possible.
- In Table 1, the authors demonstrate that adding additional exemplars can help achieve better results. It would be great to see visual results where multiple pairs assisted in improving the results.


**Questions:**

**General Comments:**
- The idea of the cosine direction to represent the editing direction has been used previously (e.g., StyleCLIP, StyleGAN-NADA). These works should be cited when introducing the CLIP-direction loss (Equation 3).

**Questions:**
- Would starting from an initial prompt depicting the desired edit speed up convergence? Similarly, does the optimization process converge faster if we provide more pairs?
- What is the significance of using InstructPix2Pix as the network? Would simply using the original Stable Diffusion model still lead to good editing results? This would be helpful to see whether the proposed scheme is robust to different diffusion models.
- Is there any significance to the “tokens” learned in the optimized conditioning text embedding? That is, is it possible to interpret the optimized token in human-understandable tokens? E.g., does quantizing the learned embeddings to real tokens lead to a coherent text prompt?
- How does changing the number of learnable tokens affect the result? How was the choice of using 10 tokens made? It would be very useful to further analyze this design choice with an ablation study.
- For the quantitative evaluations (Section 4.3), the authors specify that 300 editing directions were sampled from the clean-instructpix2pix dataset. Were these prompts used in the training of InstructPix2Pix?


**Limitations:**

The authors discuss several limitations of their method including the reliance on the diffusion model and the ability to perform small edits over the image.

---

> ### Author Rebuttal · Authors · 2023-08-08
>
> Thank you for your thoughtful feedback! We address your questions/concerns below.
>
> **It appears that the proposed method is more effective on style-based edits, as I could not find any examples to require a large change in the structure of the object. [...]**
>
> Edits that “require a large change in the structure of the object” are typically more challenging than style-based edits.
> We do have some examples that might answer your question:
> - Supplementary (Figure 4, Last row): Add a gun, add sunflowers, add roses.
> - Supplementary (index.html file, Case ID: 0282472): Add a herd of deer
> - Supplementary (index.html file, Case ID: 0135140): Add a thunderstorm
> - Supplementary (index.html file, Case ID: 0225198): Add a tiara
> - … and more.
>
> **[...] a single example may not generalize well to all different test sets. This is expected, but I am wondering how sensitive the method is to the choice of the exemplar pair. [...] one possible way to assess this could be evaluating the success rate of a single input pair across a set of test images. [...]**
>
> This is a very interesting question and suggestion. We have performed an initial investigation, but do not see any clear pattern yet. We will continue to look into this and report any interesting findings in the final version.
>
> **In Table 1, the authors demonstrate that adding additional examples can help achieve better results. It would be great to see visual results [...].**
>
> Thank you for your suggestion! We have created a Figure for this, please refer to Additional PDF File (Figure 2). We will also add this Figure into our Supplementary material.
>
> **Would starting from an initial prompt depicting the desired edit speed up convergence? Similarly, does the optimization process converge faster if we provide more pairs?**
>
> We find that initialization from the ground-truth prompt hurts performance (Table 1, Row 4-8).
> We hypothesize this may be because human-language can be mismatched with machine-preference [1].
>
> More example pairs will help to learn the desired edits faster and more precisely. Consider the settings where we have fixed all hyper-parameters and vary only the number of examples. The results (Table 1, Rows 4-8) indicate that under the same optimization steps (N =1000 steps), having more examples (1 -> 4) boosts CLIP Directional Similarity scores (0.113 -> 0.133). These scores indicate that the learned instruction is more aligned with the desired edit.
>
> **What is the significance of using InstructPix2Pix as the network? [...]**
>
> InstructPix2Pix [4] directly builds upon a pretrained Stable Diffusion model’s vast text-to-image generation capabilities, while further finetuning it with 450,000 (text instruction, before image, after image) triplets. Thus, its learned instruction space is already rich enough to cover many image-to-image translation edits, and is therefore a good starting point for our approach.
>
> Another reason is that the architecture of InstructPix2Pix is a natural fit for our task. Exploring how to make our approach work for general text-to-image models (e.g. Stable Diffusion) would be a good future research direction.
>
> **Is there any significance to the “tokens” learned in the optimized conditioning text embedding? That is, is it possible to interpret the optimized token in human-understandable tokens? [...]**
>
> This is an interesting question!
> We have tried to convert optimized tokens into natural tokens, however, we find that the natural tokens might not reflect the edit.
>
> Inspired by [3], we mapped each learned tokens to their nearest tokens within the [CLIP vocabulary corpus](https://huggingface.co/timbrooks/instruct-pix2pix/raw/main/tokenizer/vocab.json).
> For example, the human-understandable tokens in Figure 4 (Row 4, Column 2) is `ius souri anemone beans required throat alise eee`. These tokens are unrelated to the desired edit, which should be something similar to "Turn it into a watercolor painting."
>
> Our findings align with previous empirical evidence [2], which suggests a disconnect between continuous prompts and their discrete interpretations. In [2], authors demonstrated that an accurate continuous prompt tailored for a specific task (e.g., describing a precise painting) could be projected into arbitrary or even irrelevant text/statements (e.g., code or a question).
>
> **How does changing the number of learnable tokens affect the result? How was the choice of using 10 tokens made? [..]**
>
> We employ [3] to initialize tokens (Section 5 - Initialization). In [3], 8 tokens are shown to be sufficient enough to yield a stable performance. Plus <|startoftext|> and <|endoftext|> tokens, we have 10 tokens as initialization.
> We find that longer prompts do not necessarily produce better results (As demonstrated in Section 4.2 of [3]).
>
> **[...] Were 300 editing directions used in the training of InstructPix2Pix?**
>
> Yes, as the Clean-InstructPix2Pix dataset [4] does not come with a train/ test split.
> However, there are many in-the-wild examples: All images in Figure 2 and Figure 4 are in-the-wild photos (not in InstructPix2Pix dataset).
> Last row of Figure 3 is also in-the-wild. You can find further in-the-wild examples in the Supplementary (all Figure 1, Figure 2 (Row 1, 3-7), all Figure 4).
>
> **The idea of the cosine direction to represent the editing direction has been used previously (e.g., StyleCLIP, StyleGAN-NADA). These works should be cited when introducing the CLIP-direction loss (Equation 3).**
>
> Thank you for pointing it out. We have updated our paper accordingly.
>
> *Reference:*
>
> [1] Yaru et al., *Optimizing Prompts for Text-to-Image Generation*, arXiv 2022.
>
> [2] Daniel et al., *Prompt Waywardness: The Curious Case of Discretized Interpretation of Continuous Prompts*, ACL 2022.
>
> [3] Yuxin et al., *Hard Prompts Made Easy: Gradient-Based Discrete Optimization for Prompt Tuning and Discovery*, arXiv 2023.
>
> [4] Tim et al., *InstructPix2Pix Learning to Follow Image Editing Instructions*, CVPR 2023.

---

> > ### Comment · Area_Chair_RsBK · 2023-08-15
> > **Please check the author's responses**
> >
> > Dear Reviewer G8X8,
> >
> > Could you go over the authors' responses, as well as the questions raised by the other reviewers?
> >
> > Particularly, do you agree with the issues raised by the other reviewers? Do the authors' responses address the questions? Do you have any further questions for the authors?
> >
> > Thanks, Your AC

---

> > > ### Comment · Reviewer_G8X8 · 2023-08-16
> > >
> > > I appreciate the response provided by the authors that helped clarify my main reservations. I particularly appreciate the addition of the Figure showing the effect of adding more examples, which I believe will be a nice addition to the submission.
> > > I would therefore like to keep my original rating of Accept.

---

> > > > ### Author Response · Authors · 2023-08-21
> > > >
> > > > Thank you for your thoughtful review and discussion!

---

### Official Review · Reviewer_vo15 · 2023-06-28

**Soundness:** 2 fair
**Presentation:** 3 good
**Contribution:** 2 fair
**Rating:** 5
**Confidence:** 5

**Summary:**

This paper proposes a method for image editing via visual prompting. Given pairs of example that represent the “before”and “after” images of an edit, this framework can learn a text-based editing direction that can perform the same edit on the new images. Experimental results show the effectiveness of the proposed approach, even with one example pair.

**Strengths:**

The proposed framework can learn an edit direction that can be applied on new images, even with one example pair.

**Weaknesses:**

1.	The results in the first row of Fig. 3 have some artifacts in the hair and face; the results in the third row of Fig. 3 have the undesired changes in the background. Thus, the performance with only one example pair is not very good. The visual results in Fig. 4 are also not satisfactory. There should be a visual user study.


2.	How to set the values of $\lambda_{mse}$ and $\lambda_{clip}$ for different example pairs? I wonder the robustness of the proposed methods towards different hyper-parameters. Moreover, The CLIP space cannot describe all texture changes, and we should consider the balance between the reconstruction and the CLIP guidance.


3.	This framework is built based on InstructPix2Pix, which is trained with a number of “before” and “after” pairs. Thus, I wonder the performance when an example pair not in InstructPix2Pix’s training dataset is set as the condition.

**Questions:**

Please answer the questions in the weakness section.

**Limitations:**

The results of this paper is not sufficient, and more analysis about the parameters can further improve the contributions of this paper.

---

> ### Author Rebuttal · Authors · 2023-08-08
>
> Thank you for your constructive feedback! We address your questions/concerns below.
>
> **The results in the first row of Fig. 3 have some artifacts in the hair and face; the results in the third row of Fig. 3 have the undesired changes in the background.**
>
> We admit that the background can sometimes change (as in the Fig. 3 wolf -> dog result).
> You can notice that other baselines have background changes as well (Figure 3, Row 3, Column 5-7).
> We find that our results can be improved by more carefully selecting the noise.
> How to automatically select better noise is an ongoing research direction [1,2]. (Please refer to Rebuttal PDF file, Figure 3 for more qualitative result).
>
> It is worth noting that visual prompting is demonstrated to be sensitive to example pairs too [3,4]. In Figure 3 (Row 3, Column 1-2), there is snow in the background of the before-and-after pair. Thus, the learned instruction might have learned something related to “snow” to better capture the edit. We briefly discussed it in Section 6 (Discussion, Figure 7c), as there is an open question to understand what makes a good example pair.
>
> **The performance with only one example pair is not very good. The visual results in Fig. 4 are also not satisfactory. There should be a visual user study.**
>
> Without further clarification, we find the comments regarding Figure 4 difficult to respond to.
> Figure 4 demonstrates that by showing an example pair, our method can learn and replicate the distinctive characteristics of each specific art style (drawing, painting) for the same text instruction “Turn it into a drawing/ painting”.
> We perceive that the learned style is visually close to the given example pair.
>
> Regarding the visual user study: we did consider conducting a user study.
> However, we decided against it as *we felt it is not possible to guarantee a fair comparison* between our approach and the text-conditioned baselines.
>
> For example, in Figure 3 (Second row), the edit prompt is "make it a watercolor painting". Without access to an additional before-and-after image as our approach, the outputs from all baselines appear equally valid (Column 5-6), as they successfully transform the test image into a "watercolor painting". However, they may miss certain aspects that users want from the inference image such as red colored flowers in the example (Column 2).
>
> Conversely, it is also unfair to our approach (using only the before-and-after pair) as we lack knowledge of the exact edit provided by the text prompt. Using Figure 3 (Second row) as an example once more, not knowing the specific text prompt might lead to various interpretations (e.g., "colorize it", "make the coat pink", etc.). Thus it is unclear for us how to make a fair comparison (i.e., whether we should provide a reference image or instruction during the use study).
>
> **How to set the values of $\lambda_{clip}$ and $\lambda_{mse}$ for different example pairs? I wonder the robustness of the proposed methods towards different hyper-parameters. Moreover, The CLIP space cannot describe all texture changes, and we should consider the balance between the reconstruction and the CLIP guidance.**
>
> We mentioned in Section 4.1 (Line 170-173) that in all experiments, we set hyperparameters of $\lambda_{mse}$ and $\lambda_{clip}$ to 4 and 0.1, respectively. We find that this is generally sufficient to achieve good results, and that our method is not very sensitive to small changes in their exact values.
>
> We do consider the image reconstruction by Image CLIP Similarity (similar to [5]), as shown in Figure 5. Image CLIP Similarity score indicates how similar the "test" and "edited" images are. In other words, it tells us how much the edited image looks different from the test image. Results show that our method performs similarly to other state-of-the-art models.
>
> **This framework is built on InstructPix2Pix, which is trained with a number of “before” and “after” pairs. Thus, I wonder about the performance when an example pair not in InstructPix2Pix’s training dataset is set as the condition.**
>
> We indeed have many in-the-wild examples!
>
> All images in Figure 2 and Figure 4 are in-the-wild photos (not in InstructPix2Pix dataset). Last row of Figure 3 is also in-the-wild. You can find more in-the-wild examples in the Supplementary (Figure 1, Figure 2 (Row 1, 3-7), all Figure 4).
>
> *Reference:*
>
> [1] Clinton et al., *Interpolating between Images with Diffusion Models*, arXiv 2023. (Section 4.4)
>
> [2] Bahjat et al., *Imagic: Text-Based Real Image Editing with Diffusion Models*, CVPR 2023. (Supplementary - Section B)
>
> [3] Yuanhan et al., *What Makes Good Examples for Visual In-Context Learning?*, arXiv 2023.
>
> [4] Yanpeng et al., *Exploring Effective Factors for Improving Visual In-Context Learning*, arXiv 2023.
>
> [5] Tim et al., *InstructPix2Pix Learning to Follow Image Editing Instructions*, CVPR 2023.

---

> > ### Comment · Area_Chair_RsBK · 2023-08-15
> > **Please check the authors' responses**
> >
> > Dear Reviewer vo15,
> >
> > Could you go over the authors' responses, as well as the questions raised by the other reviewers?
> >
> > It seems that the authors do provide various results in the paper and supplementary material to support their method, as well as the study on the usage of the hyper-parameters. Do these convince you? Do you have any further questions for the authors?
> >
> > Thanks, Your AC

---

> > > ### Comment · Area_Chair_RsBK · 2023-08-18
> > > **[AC request] Please check the authors' responses and respond ASAP**
> > >
> > > Dear reviewer vo15,
> > >
> > > As the author-reviewer discussion period is ending soon, could you go over the authors' responses, as well as the questions raised by the other reviewers ASAP?
> > >
> > > It seems that the authors do provide various results in the paper and supplementary material to support their method, as well as the study on the usage of the hyper-parameters. Do these convince you? Do you have any further questions for the authors?
> > >
> > > Thanks, Your AC

---

> > ### Comment · Reviewer_vo15 · 2023-08-19
> >
> > Thanks for the rebuttal from the authors. After reading the rebuttal and the discussion from other reviewers, I think some of my concerns have been resolved. But I am still concerned about the quality of the results. Especially, I can not agree that the figures mentioned in fourth question are in-the-wild, I think the answers are subjective. So I decided to improve my score from borderline reject to borderline accept.

---

> > > ### Author Response · Authors · 2023-08-21
> > >
> > > Thank you for your constructive feedback!
> > >
> > > Regarding the Figures mentioned in the fourth question: We meant that they are "in-the-wild" as they do not belong to the InstructPix2Pix's training data (as originally asked by the reviewer). Specifically:
> > > * (1) The before images in Figure 2, Figure 3 (Last row), Figure 4 (Row 1-3), are randomly collected from the internet based on Google Search of random concepts (Links to original photos are provided in Supplementary, Line 78 - 88);
> > > * (2) Figure 4 (Row 4-6) are images of one of the author's dog!

---

### Official Review · Reviewer_VA7A · 2023-07-02

**Soundness:** 3 good
**Presentation:** 3 good
**Contribution:** 3 good
**Rating:** 5
**Confidence:** 4

**Summary:**

- This paper proposes a novel image editing method via visual prompts
- This paper introduces a new method to bind text-based transferring to a specific conversion between image pairs.

**Strengths:**

- The paper is well written and easy to follow.
- The motivation is clear and reasonable.
- The proposed method of using image pairs to guide text directions is novel. It introduces a new method to bind text-based transferring to a specific conversion between image pairs, which can be useful.
- The effectiveness of the method is validated.

**Weaknesses:**

- According to section 4.1, a key issue is that this work relies on existing pre-trained models to obtain high-quality paired images, I wonder will the quality heavily relied on the quality of these pre-trained models?
- The method may have limited application, since the style transfer has to be built based on existing image pairs. Which means a stable style transfer model is already available, the contribution of this work is to bind the specific transfer to a text prompt. However, during the practical application, a desired style transfer may not have available paired ground-truth images.


**Questions:**

- I wonder for one text prompt, how many pairs are required to train a stable directional prompt?

**Limitations:**

Please refer the weakness, the main limitation would be the availability of the image pairs during the real application.

---

> ### Author Rebuttal · Authors · 2023-08-08
>
> Thank you for your positive feedback! We address your questions/concerns below.
>
> **According to section 4.1, a key issue is that this work relies on existing pre-trained models to obtain high-quality paired images, I wonder will the quality heavily relied on the quality of these pre-trained models?**
>
> In practice, paired images can be provided by users. The paired examples (before-and-after) do not necessarily have to be generated by pretrained models.
>
> For example, in Figure 1 (Row 2), it is a real roadmap and satellite image. In Figure 3 (Last row), it is a real human face and photoshopped version of it. Same with Figure 7c, before-and-after images are individually collected from the internet (e.g., [this reddit link](https://www.reddit.com/r/Frozen/comments/j4afdf/elsa_anna_kristoff_in_real_life/)). Full photo attributions can be found in the last section of the Supplementary material.
> So the quality does not rely on pre-trained models.
>
> **The method may have limited application, since the style transfer has to be built based on existing image pairs. Which means a stable style transfer model is already available, the contribution of this work is to bind the specific transfer to a text prompt.**
>
> Style transfer is just one application of our method. Beyond style transfer, we can also perform other image editing tasks (ie. domain translation). For example:
> - Figure 3 (Row 3): wolf <-> dog
> - Figure 6 (Row 1): dog <-> fox
> - Figure 4 (Supplementary, Row 4): add sunflowers, add roses, etc.
> - Supplementary (index.html file, Case ID: 0282472): add deer
> - Supplementary (index.html file, Case ID: 0055735): add fog
> - Supplementary (index.html file, Case ID: 0369699: replace cliff with skyscraper
> - … and more.
>
> **During the practical application, a desired style transfer may not have available paired ground-truth images.**
>
> There are many use cases where such paired ground-truth is available.
>
> Recall our example in Introduction (Figure 1, Second row), imagine that you want to transform a roadmap image into an aerial one.  In this case, you will only have to annotate one example, and our method can help you automatically apply that transformation to new images.
>
> Another use case is *learning your specific drawing style*. Imagine that you have spent hours, days, or months to draw your cat, in your very own style. It’s very stunning, and now you want to draw again, using the same style, but now, your dog.
> Instead of spending another hours, days, or months to start all over again, you can use our method to learn your very specific style… Then a new drawing will only be minutes away!
>
> **I wonder for one text prompt, how many pairs are required to train a stable directional prompt?**
>
> Although more pairs can help (as we show in Figure 2 in the Rebuttal PDF file), in general, only one before-and-after pair is sufficient to learn the edit direction (Table 1, "Fixed noise", Row 9-12).

---

> > ### Comment · Area_Chair_RsBK · 2023-08-15
> > **Please check authors' responses**
> >
> > Dear Reviewer VA7A,
> >
> > Could you go over the authors' responses, as well as the questions raised by the other reviewers?
> >
> > Do the authors' responses about the limitation of relying image pairs convince you? Do you have any further questions for the authors?
> >
> > Thanks, Your AC

---

> > > ### Comment · Area_Chair_RsBK · 2023-08-18
> > > **[AC request] Please check authors' responses and respond ASAP**
> > >
> > > Dear reviewer VA7A,
> > >
> > > As the author-reviewer discussion period is ending soon, could you go over the authors' responses, as well as the questions raised by the other reviewers ASAP?
> > >
> > > Do the authors' responses about the limitation of relying image pairs convince you? Do you have any further questions for the authors?
> > >
> > > Thanks, Your AC

---

> > ### Comment · Reviewer_VA7A · 2023-08-21
> >
> > Thank you for the comprehensive reply. After reading the reply and other reviewers' comments, I will keep my score.

---

> > > ### Author Response · Authors · 2023-08-21
> > >
> > > Thank you for your time and effort in reviewing both our manuscript and rebuttal.

---

### Official Review · Reviewer_h5yQ · 2023-07-07

**Soundness:** 3 good
**Presentation:** 3 good
**Contribution:** 2 fair
**Rating:** 3
**Confidence:** 4

**Summary:**

This paper investigates image editing via visual prompting useful when textual descriptions cannot describe desired edits. The proposed framework inverts visual prompts into editing instructions and learns directions in the text space of the pretrained instruct pix-to-pix model. This edit direction is learned from a pair of query and target images to generate desired image editing. Results suggest, one example is sufficient to learn such directions.

**Strengths:**

- It is fascinating to see how visual cues can be translated into text-based editing directions, especially when the edits are difficult to convey through written instructions alone.
- The editing approach that is learned can be used to modify new test images with impressive accuracy. It is impressive that comparable results can be achieved even with just one example pair.
- Practical insights into image editing with diffusion models, including the potential to apply the same noise schedule for both training and testing is also interesting

**Weaknesses:**

The paper has a significant weakness in that it fails to acknowledge seminal work on image analogies, specifically, the research conducted by Hertzmann et al. (2001) and their deep learning adaptation, deep image analogies (Liao et al., 2017). While the proposed approach shares similarities with the concept of image analogies, the use of instruct pix to pix with a similar analogy lacks novelty, except for the editing directions found in textual embedding space.

Furthermore, the paper neglects to mention other related works such as SINE: SINgle Image Editing with Text-to-Image Diffusion Models (CVPR 2023) which employs style transfer for image editing through a diffusion model as seen in Figure 11 and Section 4.4 of their paper similar to image prompting. Additionally, MIDMs: Matching Interleaved Diffusion Models for Exemplar-based Image Translation (AAAI 2023) is also absent in the current work. It is crucial to compare and discuss these related works for a comprehensive analysis, which is currently missing in the manuscript.


**Questions:**

Is it possible to enhance the quality of editing directions by providing more examples (before and after images)? Also, what is the average time required to find a single editing direction?

**Limitations:**

The paper needs to properly acknowledge previous work on image analogies and lacks originality in its approach. Other related works should be mentioned and compared for a comprehensive analysis, which is currently lacking.

---

> ### Author Rebuttal · Authors · 2023-08-08
>
> Thank you for finding our paper "fascinating" and "practical". We answer your questions/concerns as below:
>
> **The paper has a significant weakness in that it fails to acknowledge seminal work on image analogies, specifically, the research conducted by Hertzmann et al. (2001) and their deep learning adaptation, deep image analogies (Liao et al., 2017)... Other related works should be mentioned and compared for a comprehensive analysis, which is currently lacking.**
>
> Thank you for your suggestion.
> We agree that *the seminal image analogies paper should have been cited*.
> We will include Image Analogies [1], Deep Image Analogies [2], and other related works in the Related Works section on Visual Prompting (Line 87-99).
>
> However, *we do not think that we lack comprehensive analysis*.
>
> We identify our work as being in the intersection of Visual Prompting (1) and Image Editing (2).
> Thus, we conducted extensive comparisons against state-of-the-art frameworks in these two domains:
>
> - (1) Visual Prompting: Our work is inspired by Visual Prompting via Image Inpainting (NeurIPS 2022) [3].
> To some extent, we believe that this work can be viewed as a modern deep learning variant of Image Analogies.
> So comparing to Visual Prompting can be represented as comparing to one of the most recent state-of-the-art work in Image Analogies.
>
> - (2) Image Editing: We conducted quantitative and qualitative experiments on a handful of state-of-the-art baselines in image editing (e.g. InstructPix2Pix [4], SDEdit [5], Imagic [6], Null-text Inversion [7]).
>
> Furthermore, we also presented comparisons between ours and Textual Inversion method [8], which can be found in the Supplementary material (Section A).
>
> **While the proposed approach shares similarities with the concept of image analogies, the use of instruct pix to pix with a similar analogy lacks novelty, except for the editing directions found in textual embedding space. The paper lacks originality in its approach.**
>
> We respectfully disagree (and we believe that other reviewers disagree too; e.g. [#VA7A](https://openreview.net/forum?id=l9BsCh8ikK&noteId=dVSnD0IySj_): "proposed method [...] is novel").
>
> It is true that we build upon InstructPix2Pix [4], but InstructPix2Pix does not support visual prompting (it seems the reviewer also acknowledges this, per "It is fascinating to see how visual cues can be translated into text-based editing directions"?).
> We have proposed a novel approach to enable image editing through visual prompting.
>
> Regarding your comment “except for the editing directions found in textual embedding space”: we actually see this is a major contribution, as we are the first to do so, and empirically demonstrate it can lead to clear advantages in image editing especially when the desired edit is difficult to describe with language.
>
> **Furthermore, the paper neglects to mention other related works such as SINE: SINgle Image Editing with Text-to-Image Diffusion Models (CVPR 2023) which employs style transfer for image editing through a diffusion model as seen in Figure 11 and Section 4.4 of their paper similar to image prompting. (Additionally, MIDMs: Matching Interleaved Diffusion Models for Exemplar-based Image Translation (AAAI 2023) is also absent in the current work. It is crucial to compare and discuss these related works for a comprehensive analysis, which is currently missing in the manuscript.)**
>
> We are happy to include these papers in the Related Work section, but note that they are only loosely related and not closely related to our work.
>
> The setting in Section 4.4 of SINE [9] is different from our visual prompting setting. Recall that we learn an edit from a before-and-after pair, then apply it to a test image. In contrast, in Section 4.4 of SINE, there is only a test image and a reference image (style transfer based on reference image). Thus, our setting is more general than SINE. Moreover, SINE requires finetuning the diffusion model again for each edit, while ours only needs to optimize one embedding vector.
>
> Likewise, our work is also only loosely similar to the MIDMs [10] setting. MIDMs is designed for exemplar-based image translation, i.e., producing image $I_{xy}$ by combining content $I_x$ and style $I_y$. In contrast, our method aims to learn edits from before-and-after images (i.e., learning the transformation from $I_x$ to $I_y$, and then applying that learned transformation to another test image). Moreover, MIDMs framework is implemented in latent space (noise space), while ours is in text-space.
>
> **What if more examples are provided?**
>
> More examples improve the performance, as depicted in Table 1 (Last 4 rows). Result indicates that increasing the number of example pairs (1 -> 4) increases the Directional CLIP score (0.113 -> 0.133).
>
> **What is the average time required to find a single editing direction?**
>
> It takes roughly 7 minutes to find an editing direction. More implementation details can be found in the Supplementary material (Section C1).
>
> *Reference:*
>
> [1] Hertzmann et al., *Image analogies*, SIGGRAPH 2001.
>
> [2] Liao et al., *Visual attribute transfer through deep image analogy*, SIGGRAPH 2017.
>
> [3] Amir et al., *Visual Prompting via Image Editing*, NeurIPS 2022.
>
> [4] Tim et al., *InstructPix2Pix Learning to Follow Image Editing Instructions*, CVPR 2023.
>
> [5] Chenlin et al., *SDEdit: Guided Image Synthesis and Editing with Stochastic Differential Equations*, ICLR 2022.
>
> [6] Bahjat et al., *Imagic: Text-Based Real Image Editing with Diffusion Models*, CVPR 2023.
>
> [7] Ron et al., *Null-text Inversion for Editing Real Images using Guided Diffusion Models*, CVPR 2023.
>
> [8] Rinon et al., *An Image is Worth One Word: Personalizing Text-to-Image Generation using Textual Inversion*, ICLR 2023.
>
> [9] Zhixing et al., *SINE: SINgle Image Editing with Text-to-Image Diffusion Models*, CVPR 2023.
>
> [10] Junyong et al., *MIDMs: Matching Interleaved Diffusion Models for Exemplar-based Image Translation*, AAAI 2023.

---

> > ### Comment · Area_Chair_RsBK · 2023-08-15
> > **Please check authors' responses**
> >
> > Dear Reviewer h5yQ,
> >
> > Could you go over the authors' responses, as well as the questions raised by the other reviewers?
> >
> > Do the authors' responses convinces you, especially on the originality/novelty part? Do you have any further questions for the authors?
> >
> > Thanks,
> > Your AC

---

> > > ### Comment · Reviewer_h5yQ · 2023-08-16
> > > **Response to Author Rebuttal**
> > >
> > > Thank you for taking the time to address the concerns raised in the review. While I appreciate your efforts in providing clarifications, there are still some areas that require further elaboration and evidence. Please find my comments below:
> > >
> > > > "To some extent, we believe that this work can be viewed as a modern deep learning variant of Image Analogies. So comparing to Visual Prompting can be represented as comparing to one of the most recent state-of-the-art work in Image Analogies."
> > >
> > > I would kindly request empirical evidence to support this claim, especially since Visual Prompting is tailored for a distinct task. A clear justification or experimental comparison would be beneficial.
> > >
> > > >  #VA7A: "proposed method [...] is novel" --
> > >
> > > The perspectives on the novelty of the proposed method may vary, it's essential to ensure that seminal works in the domain are not overlooked. A comprehensive literature review that positions your work in relation to seminal contributions would strengthen your claim on novelty that is currently missing in the paper.
> > >
> > > > "Our setting is more general than SINE"
> > >
> > > It would be beneficial to provide experimental results or concrete evidence that substantiates the claim of your paper's setting being more general than SINE. If indeed this is true, the proposed method should outperform SINE. Both the rebuttal and paper do not have experiments supporting this claim.
> > >
> > > > "MIDMs is designed for exemplar-based image translation"
> > >
> > > Agreed, and I appreciate this acknowledgment. Given they are related, a deeper exploration or comparison might improve the comprehensiveness of your work. This comparison is currently missing.
> > >
> > > > "MIDMs framework is implemented in latent space (noise space), while ours is in text-space"
> > >
> > > Given this distinction, it becomes even more crucial to draw a comparative analysis between the two. Such a comparison would offer insights into the relative advantages and potential limitations of each approach.
> > >
> > > I hope these additional comments help in refining the paper further. I look forward to seeing a more detailed and evidence-backed response.

---

> > > > ### Author Response · Authors · 2023-08-17
> > > > **Response to Reviewer h5yQ**
> > > >
> > > > The focus of our work is *a framework for converting visual prompts into editing instructions for text-to-image diffusion models*.
> > > >
> > > > Our setting is the visual-prompt setting, in which we aim to learn an edit given a few visual examples as demonstrations.
> > > >
> > > > Our work is inspired by Visual Prompting via Image Inpainting (NeurIPS 2022) [1], and it is in line with the series of papers on visual in-context learning (Section 2 - Related Work, Line 87-99).
> > > >
> > > > While we agree that Image Analogies should have been cited, we do not think that this deprecates the novelty and significance of our work.
> > > >
> > > > **"> [...] So comparing to Visual Prompting can be represented as comparing to one of the most recent state-of-the-art work in Image Analogies."
> > > > I would kindly request empirical evidence to support this claim, especially since Visual Prompting is tailored for a distinct task. A clear justification or experimental comparison would be beneficial.**
> > > >
> > > > Regarding your statement, “... especially since Visual Prompting is tailored for a distinct task” – we are a bit confused about this statement, since Visual Prompting [1] is trying to *“adapt a pre-trained visual model to novel downstream tasks without task-specific finetuning or any model modification”* and *“demonstrate[s] results on various downstream image-to-image tasks, including foreground segmentation, single object detection, colorization, edge detection, etc.”* (quotes directly taken from the Visual Prompting [1]).
> > > >
> > > > Fundamentally, both the Image Analogies paper and Visual Prompting paper are trying to solve an `A:B = C:?` analogy task, which is why we stated that the Visual Prompting approach “can be viewed as a modern deep learning variant of Image Analogies”.
> > > > We are not sure how the reviewer is expecting us to empirically support this claim. Could you please clarify?
> > > >
> > > > In case the reviewer is asking us to provide empirical comparisons to Visual Prompting, please refer to Section 4.2 (Figure 3).
> > > > Qualitative results indicate that Visual Prompting [1] (last column) is ineffective for performing image editing tasks.
> > > >
> > > > Also, as stated in the rebuttal, “We identify our work as being at the intersection of Visual Prompting (1) and Image Editing (2). Thus, we conducted extensive comparisons against state-of-the-art frameworks in these two domains: (1) Visual Prompting and (2) Image Editing.” Details can be found in Section 4 - Evaluation.
> > > >
> > > > Finally, the main focus of our work is image editing, thus we did not report other downstream tasks (e.g., image segmentation), which is beyond our scope. Nonetheless, we did briefly mention the potential of using our approach for image segmentation in Section 6 (Figure 7b) to encourage future work.
> > > >
> > > > **> #VA7A: "proposed method [...] is novel" --
> > > > The perspectives on the novelty of the proposed method may vary, it's essential to ensure that seminal works in the domain are not overlooked. A comprehensive literature review that positions your work in relation to seminal contributions would strengthen your claim on novelty that is currently missing in the paper.**
> > > >
> > > > To clarify, that specific statement in our rebuttal was a response to the reviewer’s statement:
> > > > > While the proposed approach shares similarities with the concept of image analogies, the use of instruct pix to pix with a similar analogy lacks novelty, except for the editing directions found in textual embedding space. The paper lacks originality in its approach.
> > > >
> > > > We fully agree with the reviewer that seminal works should not be overlooked, and as promised in our rebuttal, we will ensure that we properly cite the appropriate Image Analogies papers.
> > > >
> > > > However, what *we respectfully disagree with is the “lacks originality” statement*.
> > > >
> > > > The novelty aspect of our work lies in the fact that we introduce a novel "framework for converting visual prompts into editing instructions for text-to-image diffusion models.”
> > > >
> > > > We quote some statements from the other reviewers that we think support this view:
> > > > - #VA7A: “The proposed method of using image pairs to guide text directions is novel. It introduces a new method to bind text-based transferring to a specific conversion between image pairs, which can be useful.”
> > > > - #G8X8: “The authors present a simple, yet effective technique for extracting text-based editing directions in a pretrained diffusion model. This is particularly useful when a desired edit is difficult to describe with language.”
> > > > - #uc51: “It also leverages a pretrained model and only requires optimizing a vector, which makes it more appealing in applications.”
> > > >
> > > > Finally, in Section 2 - Related Work (Lines 56-99), we provided a detailed literature review and outlined the identification of our work in relation to (1) Text-to-image Models, (2) Image Editing, (3) Prompt Tuning, and (4) Visual Prompting.
> > > > In particular, our work is inspired by [1], and it is in line with the series of visual in-context learning papers (Section 2 - Related Work, Line 87-99).

---

> > > > > ### Author Response · Authors · 2023-08-17
> > > > > **Response to Reviewer h5yQ (continue)**
> > > > >
> > > > > **> "Our setting is more general than SINE"
> > > > > It would be beneficial to provide experimental results or concrete evidence that substantiates the claim of your paper's setting being more general than SINE. [...]**
> > > > >
> > > > > As stated in the rebuttal: *“The setting in Section 4.4 of SINE is different from our visual prompting setting.”*
> > > > >
> > > > > Thus, we did not perform additional experiments against SINE in both the rebuttal and our main paper. Specifically:
> > > > > * **Our setting**: We learn an edit from a before-and-after pair, then apply it to a test image.
> > > > > *For example*:
> > > > > Given before-and-after pair: Before = {A photo of a wolf}; After = {A photo of a dog}.
> > > > > We learn text-based editing direction c_T ≈ “turn a wolf into a dog”.
> > > > > Then, we can apply this learned edit to the new wolf image to get a corresponding dog image (Figure 3, Row 3).
> > > > >
> > > > > * **SINE setting** [2] (Sec 4.4, Fig. 11c): A test image and a reference image are provided, and the model performs style transfer based on the reference image.
> > > > > As stated in SINE: “Our method can generate images with the content from one and style from the other and achieve stylized generation.”
> > > > > *For example*: Input = {A photo of a dog}, Reference = {A painting of Van Gogh}.
> > > > > Output: A photo of the same dog in Van Gogh’s painting style.
> > > > >
> > > > > By saying “Our setting is more general than SINE” we mean the functionality of ours is more general than it.
> > > > >
> > > > > SINE supports (1) language instruction (e.g., Figure 3 in SINE); or (2) reference image (as style-transfer based on reference image, Figure 11c).
> > > > >
> > > > > For us, recall that InstructPix2Pix [4] functionality is a subset of ours because we can support: (1) language instruction (this is just InstructPix2Pix); (2) before-and-after pair (or “visual prompting”, this is our main setting, e.g., Figure 3); and (3) hybrid instruction (Please see Figure 4, Supplementary).
> > > > >
> > > > > For that reason, we believe that SINE is not as general as ours.
> > > > >
> > > > > **> "MIDMs is designed for exemplar-based image translation"
> > > > > Agreed, and I appreciate this acknowledgment. Given they are related, a deeper exploration or comparison might improve the comprehensiveness of your work. This comparison is currently missing.**
> > > > >
> > > > > For full context (especially for the AC and other reviewers who may also read this thread), our previous rebuttal statement was:
> > > > >
> > > > > > Likewise, our work is also only loosely similar to the MIDMs [10] setting. MIDMs is designed for exemplar-based image translation, i.e., producing image $I_{xy}$ by combining content $I_x$ and style $I_y$. In contrast, our method aims to learn edits from before-and-after images (i.e., learning the transformation from $I_x$ to $I_y$, and then applying that learned transformation to another test image). Moreover, MIDMs framework is implemented in latent space (noise space), while ours is in text-space.
> > > > >
> > > > > We will try to rephrase this in case it was confusing to the reviewer (and we apologize if that was the case):
> > > > >
> > > > > *Our setting and the MIDM [1] setting are different.* Thus, it would not make sense to compare our approach to MIDM, as we would need to create an artificially unfair/awkward setting (for either of the approaches) in order to perform the comparison.
> > > > >
> > > > > This is because the edit direction that our approach learns from the “before->after image pair” would need to be somehow represented by a single conditioning image for MIDM. But since this is not possible to achieve, in order to compare our approach to MIDM, we would need to resort to doing one of the following:
> > > > >
> > > > > - (i) Given our `A:B = C:D` analogy setting, where A is the before reference image, B is the after reference image, C is the test image, and D is the model’s output image, we input the C and B images to MIDM, to generate the D image (i.e., here, C is the test image and B is the conditioning image). This is unfair to MIDM because it does not get to see A, and worse, it’s not capturing the desired A->B edit that our approach would learn. So the output of our model (which would be the learned A->B edit applied to C) and the output of MIDM would not be comparable.
> > > > >
> > > > > - (ii) Given our `A:B = C:D` analogy setting, do not show the A image to our approach, and instead  let it learn the C->B edit direction, and then apply that edit to C. This would be awkward for our approach because it would simply reproduce B. And this is not the same as what MIDM is trying to achieve, which is to combine the properties of B and C to produce D. So again, the outputs of the two models would not be comparable.
> > > > >
> > > > > This is why we are saying that the settings are different. Even though the two settings are different, we will include a brief discussion about it in the related work. We hope this clarifies any confusion, please let us know if there are further questions.

---

> > > > > > ### Author Response · Authors · 2023-08-17
> > > > > > **Response to Reviewer h5yQ (continue)**
> > > > > >
> > > > > > We want to re-emphasize that the works that we have already compared to in our main paper and supplementary:
> > > > > > - InstructPix2Pix [4] (CVPR 2023)
> > > > > > - Imagic [6] (CVPR 2023)
> > > > > > - Null-text Inversion [7] (CVPR 2023)
> > > > > > - Textual Inversion [8] (ICLR 2023)
> > > > > > - Visual Prompting [1] (NeurIPS 2022)
> > > > > > - SDEdit [5] (ICLR 2022)
> > > > > >
> > > > > > These works are more relevant, and we clearly showed our approach’s competitive performance with respect to those state-of-the-art related works.
> > > > > >
> > > > > > **> "MIDMs framework is implemented in latent space (noise space), while ours is in text-space"
> > > > > > Given this distinction, it becomes even more crucial to draw a comparative analysis between the two. Such a comparison would offer insights into the relative advantages and potential limitations of each approach.**
> > > > > >
> > > > > > Like mentioned above, we did not provide additional comparison to MIDMs as our setting differs.
> > > > > >
> > > > > > We indeed include comparison to a handful of image editing methods that are implemented in latent space (noise space) (e.g., InstructPix2Pix, SDEdit, Imagic). Please refer to Section 4 - Evaluation in the main paper, and Section B in the Supplementary material.
> > > > > >
> > > > > > *Reference:*
> > > > > >
> > > > > > [1] Amir et al., *Visual Prompting via Image Editing*, NeurIPS 2022.
> > > > > > [2] Zhixing et al., *SINE: SINgle Image Editing with Text-to-Image Diffusion Models*, CVPR 2023.
> > > > > > [3] Junyong et al., *MIDMs: Matching Interleaved Diffusion Models for Exemplar-based Image Translation*, AAAI 2023.
> > > > > > [4] Tim et al., *InstructPix2Pix Learning to Follow Image Editing Instructions*, CVPR 2023.
> > > > > > [5] Chenlin et al., *SDEdit: Guided Image Synthesis and Editing with Stochastic Differential Equations*, ICLR 2022.
> > > > > > [6] Bahjat et al., *Imagic: Text-Based Real Image Editing with Diffusion Models*, CVPR 2023.
> > > > > > [7] Ron et al., *Null-text Inversion for Editing Real Images using Guided Diffusion Models*, CVPR 2023.
> > > > > > [8] Rinon et al., *An Image is Worth One Word: Personalizing Text-to-Image Generation using Textual Inversion*, ICLR 2023.

---

> > > > > > > ### Comment · Reviewer_h5yQ · 2023-08-19
> > > > > > >
> > > > > > > The authors have provided detailed explanations for several points. Some areas still require further exploration, empirical evidence, and clarification. Addressing these concerns comprehensively will enhance the quality and impact of the work. Relying on positive feedback from other reviewers does not negate the concerns raised by a particular reviewer.
> > > > > > >
> > > > > > > The major concern is claims regarding Visual Prompting: The authors' reliance on the Visual Prompting paper as a primary point of comparison is evident. However, the assertion that Visual Prompting can be represented as a modern deep-learning variant of Image Analogies requires more concrete justification. While both papers may be attempting to solve an A:B = C:? analogy task, the methodologies, underlying principles, and applications are vastly different -- visual prompting is designed for downstream image-to-image tasks (segmentation, edge detection, etc) and not for image editing. Therefore, as mentioned by the authors they will not be effective for this task. In contrast to the author's claim: The visual Prompting approach *cannot* be viewed as a modern deep learning variant of Image Analogies.
> > > > > > >
> > > > > > > **Ignoring a directly related method -- Deep Image Analogies (Siggraph 2017) -- in this case is concerning.** There is also a PyTorch implementation of the same: https://github.com/Ben-Louis/Deep-Image-Analogy-PyTorch
> > > > > > >
> > > > > > > I would be inclined to increase my rating if the authors would have directly provided a comparison against the most relevant and direct baseline.

---

> > > > > > > > ### Author Response · Authors · 2023-08-20
> > > > > > > >
> > > > > > > > **The major concern is claims regarding Visual Prompting: The authors' reliance on the Visual Prompting paper as a primary point of comparison is evident. However, the assertion that Visual Prompting can be represented as a modern deep-learning variant of Image Analogies requires more concrete justification. While both papers may be attempting to solve an A:B = C:? analogy task, the methodologies, underlying principles, and applications are vastly different -- visual prompting is designed for downstream image-to-image tasks (segmentation, edge detection, etc) and not for image editing.
> > > > > > > > Therefore, as mentioned by the authors they will not be effective for this task.**
> > > > > > > >
> > > > > > > > We are a bit confused by this statement:
> > > > > > > > > -- visual prompting is designed for downstream image-to-image tasks (segmentation, edge detection, etc) and not for image editing.
> > > > > > > >
> > > > > > > > In Visual Prompting [1], the authors have shown successful applications for a wide range of computer vision tasks, *including image colorization and style transfer* ([Figure 1 in Visual Prompting](https://arxiv.org/pdf/2209.00647.pdf) [1]).
> > > > > > > > We perceive that style transfer and image colorization are examples of image editing.
> > > > > > > >
> > > > > > > > Indeed, style transfer is a main application of image analogies (e.g., [Figure 12 in Image Analogies](https://mrl.cs.nyu.edu/publications/image-analogies/analogies-72dpi.pdf) [3]).
> > > > > > > > For that reason, we believe that Visual Prompting is suitable as an image editing baseline.
> > > > > > > >
> > > > > > > > **In contrast to the author's claim: The visual Prompting approach cannot be viewed as a modern deep learning variant of Image Analogies.**
> > > > > > > >
> > > > > > > > As stated previously, we believe that Visual Prompting [1] can be viewed as a modern deep learning variant of Image Analogies, given that:
> > > > > > > >
> > > > > > > > - (1) Visual Prompting is also trying to solve an `A:B = C:?` analogy task
> > > > > > > > - (2) Visual Prompting has shown successful cases with style transfer and image colorization
> > > > > > > >
> > > > > > > > Could you please give a concrete justification for why “Visual Prompting approach cannot be viewed as a modern learning variant of Image Analogies”?
> > > > > > > >
> > > > > > > > **Ignoring a directly related method -- Deep Image Analogies (Siggraph 2017) -- in this case is concerning. There is also a PyTorch implementation of the same: https://github.com/Ben-Louis/Deep-Image-Analogy-PyTorch.**
> > > > > > > >
> > > > > > > > The setting in Deep Image Analogy [2] is different from ours.
> > > > > > > >
> > > > > > > > * Deep Image Analogy [2]: `A : ? = ? : B’`.
> > > > > > > > Quote from [2]: "Given an image pair $A$ and $B’$, which may differ in appearance but have similar semantic structure, the goal is to find the mapping from $A$ to $B’$ (or from $B’$ to $A$) for visual attribute transfer."
> > > > > > > > *Input*: Two images: $A$ and $B’$.
> > > > > > > > *Output*: Two images: $A’$ and $B$.
> > > > > > > > (As clearly stated in Section 4.6 - [Algorithm 1](https://arxiv.org/pdf/1705.01088.pdf) in [2])
> > > > > > > >
> > > > > > > > * Ours: `A : A’ = B : ?`.
> > > > > > > > We learn the transformation from $A \rightarrow A’$, then apply it to $B$.
> > > > > > > > *Input*: Two images: $A$ and $A’$.
> > > > > > > > *Output*: Editing direction $c_T \approx A \rightarrow A’$.
> > > > > > > > (As clearly stated in Section 3.4 - Algorithm 1 in our main paper)
> > > > > > > >
> > > > > > > > Thus, it would not make sense to compare our approach to Deep Image Analogy [2].
> > > > > > > >
> > > > > > > > - (i) For Deep Image Analogy [2], it needs to access image $B’$, which is not available in our setting.
> > > > > > > > - (ii) For our approach, without access to $A’$, we can not learn editing direction $A \rightarrow A’$.
> > > > > > > >
> > > > > > > > *Reference*:
> > > > > > > > [1] Amir et al., *Visual Prompting via Image Inpainting*, NeurIPS 2022.
> > > > > > > > [2] Jing et al., *Visual Attribute Transfer through Deep Image Analogy*, SIGGRAPH 2017.
> > > > > > > > [3] Hertzmann et al., *Image analogies*, SIGGRAPH 2001.

---

### Official Review · Reviewer_uc51 · 2023-07-12

**Soundness:** 3 good
**Presentation:** 3 good
**Contribution:** 3 good
**Rating:** 6
**Confidence:** 5

**Summary:**

In this paper, the authors propose a new method that can perform visual prompting via a pair of exemplar images through a pretrained text-based instruction image editing model. The method introduced in this paper only requires optimizing over the text conditioning vector in order to perform visual instruction editing. The authors have shown strong qualitative results demonstrating various application scenarios.

**Strengths:**

This model introduced in this paper solves a very interesting and important application problem that is exemplar-based visual prompting of image generative models. The paper is very well written and the results look very promising and this model can be easily directly applied in a large range of intuitive applications by daily users. It also leverages a pretrained model and only requires optimizing a vector, which makes it more appealing in applications. The authors have also performed thorough ablation study.

**Weaknesses:**

1. The quantitative evaluations are not very comprehensive. The authors only sampled 1k images to perform quantitative analysis, and did not report any fidelity scores. For editing controllability, they only report CLIP related scores, which is not very representative in many circumstances [1].
2. When explaining the method, the authors sometimes mix the prior work with their contributions. A prime example would be Section 3.2, which is redundant because it is the same as InstructPix2Pix, but sectioning in this way makes it look like a new contribution at the first glance. I would recommend the authors to make clear distinctions between their contributions and the prior literature.

[1] Tristan Thrush, Ryan Jiang, Max Bartolo, Amanpreet Singh, Adina Williams, Douwe Kiela, Candace Ross. “Winoground: Probing vision and language models for visio-linguistic compositionality”. CVPR 2022. https://arxiv.org/pdf/2204.03162.pdf


**Questions:**

1. How similar does the exemplar visual prompt and query image should look? Can the authors give some additional ablation study on the similarity between the visual prompt and the query images?
2. How long does it take to sample one image?
3. Since CLIP is not a perfect encoder and has various known limitations, I am wondering how well this method handles tasks such as color changes or duplicating objects.
4. Related to Weakness (1), I think it is also possible to report more standardized metrics such as KID (fidelity score for small number of samples), IoU and (masked) LPIPS with tasks like semantic segmentation map to image and image compositing. Can the authors report some standardized metrics for some of these tasks?
5. Why do more examples hurt the performance (according to Table 1)?

~~I am happy to raise my score if the authors can add experiments to address my concerns.~~

***After the rebuttal discussion, I would like to raise my score from 5 to 6.***

**Limitations:**

Although the imitations are extensively discussed in the paper, the discussion of potential ethical issues is missing. Since this paper uses a pretrained large image generative model, which is known to have various societal issues, I would highly recommend the authors to include a broader impact statement in their paper.

---

> ### Author Rebuttal · Authors · 2023-08-08
>
> Thanks for your positive feedback! We address your questions/concerns below.
>
> **Only sampled 1k imgs to perform quantitative analysis**
>
> We believe that ~1000 images are sufficient to validate our approach. This is in line with other related work; e.g., Imagic (CVPR 2023): ~100 image pairs; Null-text Inversion (CVPR 2023): ~1000 test images.
>
> **[...] did not report any fidelity scores**
>
> Thanks for your suggestion! Below, we present new experiment using the LPIPS score [1] for 25 in-the-wild image pairs.
> To recap, we use a **before-and-after** image pair to learn an edit, then apply it to the **test** image to get an **edited** version.
>
> We use LPIPS using two ways:
> - LPIPS(test, edited): Measures the similarity between the test and edited image. Lower score indicates that the edited image has similar image fidelity to the test image.
> - LPIPS(after, edited): Assesses how the edited image aligns with the after image. Lower score indicates that the edits successfully follow the visual cues provided by the after image.
>
> ||SDEdit [3]|Ip2p [4]|Ours|
> |---|---|---|---|
> |LPIPS (test, edited)|**31.34**|46.40| 43.73|
> |LPIPS (after, edited)|71.42|71.15|**62.52**|
>
> Results show that our model demonstrates competitive performance to state-of-the-art baselines. However, it is worth to note that while an edit may lead to significant changes, achieving lower LPIPS(test, edited) scores could indicate less meaningful edits, as the edited image closely resembles the original test image.
>
> Furthermore, in LPIPS(after, edited) score, our model scores lowest. This suggests that our edited images align more perceptually with after images compared to text-conditioned methods. Note that the baselines does not have access to the before-and-after pair – However, this precisely supports the key idea of our approach; i.e., using before-and-after images enhances intended edits which may not sufficiently be captured with text alone.
>
> It is worth highlighting that real dataset for visual prompting-based image editing is lacking. Substantial ongoing research efforts are dedicated to creating diverse and precise paired datasets [2]. As a result, we were unable to compute on a larger dataset.
>
> **Standardized metrics for tasks like semantic segmentation**
>
> As we focus on image editing, we did not report any downstream computer vision tasks (i.e., semantic segmentation, image compositing), which are beyond our scope.
>
> **How similar does the exemplar visual prompt and query image should look?**
>
> As long as the example reference images and the query image are roughly in the same domain, there are no constraints on how similar they should be.
> For example, In Fig. 3 (Last row), different people are in example and test images. In Fig. 6 (First row), backgrounds and people in example and test images fully change.
> On the other hand, for entirely different domains (e.g., dog->cat as example, landscape as query), our method will not work.
>
> **How long does it take to sample one image?**
>
> It takes 4 seconds to sample one image.
>
> **Since CLIP is not a perfect encoder [...] how well this method handles color changes or duplicating objects.**
>
> Qualitative results about color changing cases can be found in our main paper:
> - Fig. 3 (Row 3: gray wolf -> brown dog)
> - Fig. 6 (Row 2: green grass field -> brown grass field)
>
> It can also be spotted several times in our Supplementary (index.html), e.g., ID: 0207894 (red -> blue hair)
>
> For "duplicating objects" it seems like your question is: What happens when there are two visually similar objects in a test photo? Please refer to ID: 0307987 - boats -> camels (Supplementary, index.html) as an example. The results show that each boat shown in the test images will be transformed into a camel. For more results, please see ID 0258921 (cow -> giraffe), ID: 0016482 (human -> cheetah).
>
> With all the cases we have listed above, we see that our approach works reasonably well with a CLIP encoder.
>
> **Why more examples hurt the performance (Tab. 1)?**
>
> More examples improves the performance. Recall that in diffusion-based models, different noises lead to different outputs (Fig. 6). Thus, in Tab. 1 (Column 6-8, "Random noises"), quantitative results can vary a lot.
>
> However, we find that using the same noise sequence from training to test leads to more stable results (Sec. 5 - Noises). Based on this observation, we have more reliable quantitative results in Tab. 1 (Column 9-11, "Fixed noise"). In particular, more example pairs (1 -> 4) increase Directional CLIP score (0.113 -> 0.133).
>
> **[...] the authors sometimes mix the prior work with their contributions. [...] Sec. 3.2 is redundant because it is the same as InstructPix2Pix [...]**
>
> Thanks for your feedback. We do not want to cause any confusion here.
> While we use the same image reconstruction loss (MSE loss) as in InstructPix2Pix [4], our goals differ.
> -   InstructPix2Pix finetunes the model: update $\epsilon_{\theta}$, fixed $c_T$
> -   Ours optimizes the instruction: fixed $\epsilon_{\theta}$, update $c_T$ (As presented in Algorithm 1, Line 14)
>
> We will change Line 141:
> “We  ̶f̶o̶l̶l̶o̶w̶ ̶t̶h̶e̶ ̶s̶a̶m̶e̶ ̶s̶t̶r̶a̶t̶e̶g̶y̶ ̶a̶s̶ ̶[̶4̶]̶ employ a pretrained text-conditioned image editing model proposed in [4] [...]”
>
> **Although the limitations are extensively discussed [...] discussion of potential ethical issues is missing.**
>
> It is true that our method might inherit unwanted bias from diffusion models (As we briefly mentioned in Sec. 6, Line 256). We will further clarify potential ethical issues in our revision.
>
> *Reference:*
>
> [1] Richard et al., *The Unreasonable Effectiveness of Deep Features as a Perceptual Metric*, CVPR 2018
>
> [2] Kai et al., *MagicBrush: A Manually Annotated Dataset for Instruction-Guided Image Editing*, arXiv 2023
>
> [3] Chenlin et al., *SDEdit: Guided Image Synthesis and Editing with Stochastic Differential Equations*, ICLR 2022
>
> [4] Tim et al., *InstructPix2Pix Learning to Follow Image Editing Instructions*, CVPR 2023

---

> > ### Comment · Area_Chair_RsBK · 2023-08-15
> > **Please check authors' responses**
> >
> > Dear Reviewer uc51,
> >
> > Could you go over the authors' responses, as well as the questions raised by the other reviewers?
> >
> > Do the additional experiments the authors provided convince you? Do you have any further questions for the authors?
> >
> > Thanks,
> > Your AC

---

> > ### Comment · Reviewer_uc51 · 2023-08-18
> > **Thank you for your response**
> >
> > I would like to thank the authors for responding to my concerns. Most of my questions and concerns have been answered and therefore I would like to raise my score to 6. However, I do want to clarify that
> >
> > 1. While many prior works have used very few images to evaluate FID, FID is only consistent and stable when we have more than 10k samples [1].
> >
> > 2. By semantic segmentation map to image task, I meant the edits to the images can be derived from semantic segmentation maps.
> >
> > It would be interesting to see the authors to sample more images to evaluate FID and apply their method to the tasks I mentioned above in the future, but in my opinion this paper deserves an accept with the content provided so far in the paper, supplemantary materials and the rebuttal discussion.
> >
> >
> > [1] Mikołaj Bińkowski, Danica J. Sutherland, Michael Arbel, Arthur Gretton. Demystifying MMD GANs. ICLR 2018.

---

> > > ### Author Response · Authors · 2023-08-21
> > >
> > > Thank you for your clarification.
> > > We will continue to look into your suggestions and report any interesting findings in the final version.

---

### Author Rebuttal · Authors · 2023-08-08

We propose a framework for *inverting visual prompts into editing instructions* for text-to-image diffusion models.
Furthermore, our method can combine instructions between learned and natural language, *yielding a hybrid editing instruction that is more precise*.


We are grateful that **all reviewers appreciate either the originality, experimentation, clarity, and/or significance** of our paper.
- **Originality**: "very interesting and important" (#uc51); "impressive" (#h5yQ), "clear and reasonable", (#VA7A), "simple, yet effective" (#G8X8), "can learn an edit ... even with one example pair." (#vo15).
- **Experimentation**: "thorough ablation study" (#uc51), "effectiveness is validated" (#VA7A).
- **Clarity**: excellent presentation (#G8X8), "well written" (#VA7A, #uc51), good (#h5yQ, #vo15).
- **Significance**: "fascinating" and "practical" (#h5yQ), "intuitive application" (#uc51), "useful" (#VA7A), "applicable and easy to use" (#G8X8).
---

We thank the reviewers’ time and effort in reviewing our paper.
We have incorporated reviewers' feedback into our revision.

We are glad that reviewer #G8X8 finds the hybrid instruction [*“very interesting and beneficial”*](https://openreview.net/forum?id=l9BsCh8ikK&noteId=tZfjSeHAms).
We will move this application into the main paper from the supplementary material as suggested.

Answers to individual reviewers are addressed below each review. Please let us know if you have any additional questions or concerns.

---

### Decision · Program_Chairs · 2023-09-21

**Decision:**

Accept (poster)

**Comment:**

After the rebuttal, most reviewer concerns are addressed by the authors. There remains few questions:
1. If visual prompting is suitable for image editing tasks.
2. If visual prompting can be viewed as deep learning variant of image analogies
3. If it is reasonable to have a direct comparison between the proposed and the deep image analogies work.

After going over the literature and the comments made by the reviewers and the authors, the AC decide to accept this paper since
1. The debates around visual prompting vs. image analogies should not ignore the fact that this paper makes a major contribution toward image editing.
2. Practically, it is indeed hard to make a direct comparison quantitatively between the proposed and the deep image analogies work, and alternatively, the authors provide comprehensive discussion in the rebuttal.
3. The authors provide comprehensive experiments to compare with the image editing frameworks that can handle the problem setting in this paper.

Finally, the AC asks the authors to include all the discussions among the related works (in an organized way) in the revised paper.